

# Combined measurement and modeling of the hydrological impact of hydraulic redistribution using CLM4.5 at eight AmeriFlux sites

C. Fu[1], G. Wang[1], M. L. Goulden[2], R. L. Scott[3], K. Bible[4], Z. G. Cardon[5]

[1]Department of Civil and Environmental Engineering, and Center for Environmental Science and Engineering, University of
5 Connecticut, Storrs, CT 06269-3037, United States.
[2]Department of Earth System Science, University of California, Irvine, CA 92697-3100, United States.
[3]Southwest Watershed Research Center, USDA-Agricultural Research Service, Tucson, AZ 85719, United States.
[4]Wind River Canopy Crane Research Facility, School of Environmental and Forest Sciences, University of Washington,
Carson, WA 98610, United States.
[5]The Ecosystems Center, Marine Biological Laboratory, Woods Hole, MA 02543, United States.

*Correspondence to*: G. Wang (gwang@engr.uconn.edu)

**Abstract.** Effects of hydraulic redistribution (HR) on hydrological, biogeochemical, and ecological processes have been demonstrated in the field, but the current generation of standard earth system models does not include a representation of HR. Though recent studies have examined the effect of incorporating HR into land surface models, few (if any) has tackled
the magnitude of the HR flux itself or the soil moisture dynamics from which HR magnitude can be directly inferred. Here we incorporated Ryel et al.'s (2002) empirical equation describing HR into the NCAR Community Land Model Version 4.5 (CLM4.5), and examined the ability of the resulting hybrid model to capture the magnitude of HR flux and/or soil moisture dynamics from which HR can be directly inferred, to assess the impact of HR on surface water and energy budgets, and to explore how it may depend on climate regimes and vegetation conditions. Eight AmeriFlux sites characterized by contrasting
climate regimes and multiple vegetation types were studied, including the US-Wrc Wind River Crane site in Washington State, the US-SRM Santa Rita Mesquite Savanna site in southern Arizona, and six sites along the Southern California Climate Gradient (US-SCs, g, f, w, c, and d). HR flux, evapotranspiration, and soil moisture were properly simulated in the present study, even in the face of various uncertainties. Our cross-ecosystem comparison showed that the timing, magnitude, and direction (upward or downward) of HR vary across ecosystems, and incorporation of HR into CLM4.5 improved the
model-measurement match particularly during dry seasons. Our results also reveal that HR has important hydrological impact (on evapotranspiration, Bowen ratio, and soil moisture) in ecosystems that have a pronounced dry season but are not overall so dry that sparse vegetation and very low soil moisture limit HR.

## 1 Introduction

Hydraulic redistribution (HR) is the transport of water from wetter to drier soils through plant roots (Burgess et al., 1998).
Several recent reviews (Neumann and Cardon, 2012; Prieto et al., 2012; Sardans and Peñuelas, 2014) summarize results from the hundreds of empirical and modeling papers describing HR that have emerged over the last three decades.



Monitoring of sap flow, soil water potential, and soil moisture content all indicate that HR can occur in many ecosystems worldwide, ranging in climate from arid to wet, particularly if the system has a pronounced dry season. HR-induced transport of water can be upward (as "hydraulic lift") from moist deep soils to dry shallow soils (Richards and Caldwell, 1987), downward (as "hydraulic descent") following a precipitation event (Ryel et al., 2003), or lateral (Brooks et al., 2002).

Though effects of HR on hydrological (e.g. Scott et al., 2008), biogeochemical (e.g. Domec et al., 2012; Cardon et al., 2013), and ecological (e.g. Hawkins et al., 2009) processes have been amply demonstrated in the field, the current generation of standard dynamic global vegetation and earth system models do not include a representation of HR (Neumann and Cardon, 2012; Warren et al., 2015). The several modeling studies at ecosystem and regional scales that do include HR do so by incorporating empirical equations describing HR (Ryel et al., 2002) into various land surface models (Lee et al., 2005,

CAM2-CLM; Zheng and Wang, 2007, IBIS2 and CLM3; Baker et al., 2008, SiB3; Wang, 2011, CLM3; Li et al., 2012, CABLE; Yan and Dickinson, 2014, CLM4.0). For example, Li et al. (2012) modeled three evergreen broadleaf forests, found in tropical, subtropical, and temperate climate, and showed that the ability of CABLE to match observed evapotranspiration and soil moisture was improved by including HR and dynamic root water uptake. However, most of these studies focused on how including HR might improve the model performance in simulating ET and in some cases soil

moisture, and few (if any) has tackled the magnitude of the HR flux itself or the soil moisture dynamics from which HR magnitude can be directly inferred. It is not clear from these previous studies whether the HR-derived model performance might be caused by HR compensating for other hydrological deficiencies in the default model.  In this study, we attempt to address this research gap based on both field measurements and numerical modeling at an ecologically broad selection of eight AmeriFlux sites characterized by contrasting climate regimes and multiple vegetation types.  Of the eight sites, two

have a long history of empirical research focused on HR: the US-Wrc Wind River Crane site in the Pacific Northwest (Washington state), and the US-SRM Santa Rita Mesquite Savanna site in southern Arizona. The other six are new sites along the Southern California Climate Gradient (US-SCs, g, f, w, c, and d), each with a pronounced dry season, where we suspect HR may occur during dry periods.

At one of the six Southern CA Climate Gradient sites (the James Reserve, US-SCf), Kitajima et al. (2013) recently used

the HYDRUS-1D model and isotopic measurements of xylem water to show trees and shrubs use deep water, probably delivered both by HR and to some extent by capillary rise,  during summer drought. In the Pacific Northwest, adjacent to the Wind River Canopy Crane Research Facility (US-Wrc), stands of Douglas fir (Pseudotsuga menziesii (Mirb.) Franco) have been the focus of numerous papers examining the importance of HR in this overall-moist but seasonally-dry ecosystem. For example, Brooks et al. (2002) used sap flow and soil moisture information to show that 35% of the total day-time water

consumption from the upper 2m of soil was replaced by HR during July-August in 2000. Brooks et al. (2006) further reported that HR was negligible in early summer but increased to 0.17 mm/d by late August. Meinzer et al. (2004) reported that the seasonal decline of soil water potential was greatly reduced by HR. Based on monitoring of sap flow of Prosopis velutina Woot (velvet mesquite) and soil moisture, both hydraulic lift and hydraulic descent were found at (Scott et al., 2008) or near (Hultine et al., 2004) the Santa Rita Arizona savanna (US-SRM) site.





The objective of this study is to examine the performance of a commonly used modeling approach, the Ryel et al. (2002) approach, in capturing the magnitude of HR flux and/or soil moisture dynamics from which HR can be directly inferred, to demonstrate the impact of HR on surface water and energy budgets, and to explore how it may depend on climate regimes and vegetation conditions. This is done through incorporating Ryel et al.'s (2002) simple empirical equation for HR flux into the NCAR Community Land Model Version 4.5, and applying the hybrid model to the eight AmeriFlux sites characterized by contrasting climate regimes and multiple vegetation types.

## 2 Materials and Methods

### 2.1 Study sites

The sites in this study were chosen based on several criteria. Concurrent meteorological forcing data, soil moisture data throughout the soil profile, and evapotranspiration (ET) data for a continuous period of several years had to be available. The sites cover a range of annual rainfall amounts and vegetation types, and have seasonally dry climate - a good indicator of ecosystems where HR may occur (Neumann and Cardon, 2013). Two of the eight sites (US-SRM and US-Wrc) were specifically chosen because they have a strong record of hydraulic redistribution research. In contrast, the six Southern California gradient sites were chosen because it was not yet known whether HR occurred at them, and modeling results could be compared to new empirical data. Table 1 presents location, elevation, climate, vegetation type, annual precipitation, average temperature, and years for which we have atmospheric forcing data, for each of the eight Ameriflux sites. Further details about these eight sites can be found on the AmeriFlux website (http://ameriflux.lbl.gov/sites/site-search/). All sites except Santa Rita Mesquite have a Mediterranean climate (rainy winters, dry summers); Santa Rita Mesquite (US-SRM) is a semi-arid site with a dominant summer rainy season.   Precipitation varies from ~2200 mm (US-Wrc) to ~100 mm (US-SCw) per year. Average temperature ranges from 8.7 (US-Wrc) to 23.8 degrees C (Sonoran Desert US-SCd). Vegetation ranges from needle-leaf and broad leaf forest to chaparral, grassland, and desert perennials and annuals.

### 2.2 CLM4.5 parameterization

The NCAR Community Land Model Version 4.5 (CLM4.5) (Oleson et al., 2013) is used in this study to simulate the energy fluxes and hydrological processes at the eight AmeriFlux sites. Table 2 presents the sources of data used as model input, for atmospheric forcing and surface properties including coverage of different plant functional types (PFTs), LAI, canopy height, soil texture, and soil organic matter content. At each site, atmospheric forcing data used to drive CLM4.5 are taken from the corresponding AmeriFlux tower.  Surface properties in the model are set to reflect the AmeriFlux site conditions when such information is available and were drawn by interpolation from corresponding gridded datasets in the NCAR database (Oleson et al., 2013, and Notes S1) in the absence of site-specific data.

Within CLM4.5, the Clapp and Hornberger "$B$" parameter (the exponent in the soil water retention curve that varies substantially with soil texture) strongly influences simulated soil moisture.  We used available sources of soil texture



information for the eight sites (Table 2) to set the range of appropriate "*B*" for each site and depth (Table 3), following Clapp and Hornberger's (1978) ranges of "*B*" for different soil types. Within each range, however, we tuned the values for "*B*" with depth to get a good match between modeled and measured soil moisture.

The atmospheric forcing data at the US-Wrc and US-SRM sites include incident long-wave radiation, incident solar
radiation, precipitation, surface pressure, relative humidity, surface air temperature, and wind speed. Because incident long-wave radiation and surface pressure data were not available at the six Southern California sites, CLM4.5 assumes standard atmospheric pressure and calculates the incident long-wave radiation based on air temperature, surface pressure, and relative humidity (Idso, 1981). Gap-filled atmospheric forcing data are at 30-minute resolution, and the time step for model simulations is also 30 minutes. Time frames for which atmospheric forcing data are available for each site are shown in
Table 1.

## 2.3 HR model parameterization

To quantify HR, we incorporated Ryel et al.'s (2002) equation into CLM4.5. This equation has been widely used in HR modeling studies (Lee et al., 2005; Zheng and Wang, 2007; Wang, 2011; Li et al., 2012) and its variations (e.g. Yu and D'Odorico, 2015). HR-induced soil water flux $q_{HR}(i, j)$ (cm h$^{-1}$) between a receiving soil layer $i$ and a giving soil layer $j$ is
quantified as:

$$q_{HR}(i,j) = -C_{RT}\Delta\varphi_m c_j \frac{F_{root}(i)F_{root}(j)}{1-F_{root}(j)} \qquad (1)$$

By summing all giving and receiving layer pairs within the soil column, total $q_{HR}$ can be calculated. $C_{RT}$ is the maximum radial soil-root conductance (cm MPa$^{-1}$ h$^{-1}$), $\Delta\varphi_m$ is water potential difference between two soil layers (MPa), $F_{root}(i)$ is root
fraction in layer $i$ (weighted average of PFT-level root fractions; Zeng, 2001), and the factor reducing soil-root conductance for water in the giving layer $c_j$ is

$$c_j = \frac{1}{1+(\frac{\varphi_j}{\varphi_{50}})^b} \qquad (2)$$

In equation (2), $\varphi_j$ is soil water potential in layer $j$ (MPa), $\varphi_{50}$ is the soil water potential where soil-root conductance is
reduced by 50% (MPa), and $b$ is an empirical constant. The relationship between root hydraulic conductivities and soil moisture in equation (2) is similar to that in Amenu and Kumar (2008). Values for b (3.22) and $\varphi_{50}$ (-1 MPa) were taken from Ryel et al. (2002) for lack of site-specific parameters, and we tested the model sensitivity to the parameters $C_{RT}$, $b$, and $\varphi_{50}$ at each site. Rather than tuning $C_{RT}$ as Ryel et al. (2002) did to match modeled HR (calculated in equation 1) to measured HR (from soil sensor data) after a saturating rain, we based the tuning of $C_{RT}$ on comparison of modeled and measured
magnitude and dynamics of water content in upper soil layers (0-30 cm) at hourly scale during dry periods. At the three drier southern CA sites (US-SCw, US-SCc, and US-SCd), $C_{RT}$ was further adjusted to relatively small values (0.05-0.1) to limit the hydraulic descent in order to reduce the model bias for soil water potential during dry periods. The modeled soil





water potential would be always higher than -1 MPb during dry periods if $C_{RT} > 0.1$, which is not realistic for such dry sites. Specific values of the parameters in the "Ryel et al. 2002" equation used for the eight study sites are shown in Table 4.

## 2.4 Combined model

Two multi-year simulations were carried out at each of the eight study sites. "Without HR" used the default land surface model CLM4.5; "with HR" (CLM4.5+HR) used the version of the model including Ryel's representation of HR. To distinguish the influences of "$B$" and HR on the soil moisture modeling, the tuning of the parameter "$B$" was done based on wet season (with high soil moisture) when the HR influence is negligible at the US-Wrc and CA sites. Therefore the "$B$" values do not depend on whether the tuning was done with CLM4.5 or with CLM4.5+HR. At the SRM site, HR is mainly in the form of hydraulic descent during rainfall events (as shown later in the Results section), we tuned "$B$" during dry periods when hydraulic descent was minimum to make the minimum value of the modeled soil moisture from CLM4.5 be close to the observation for surface soil layers. The "$B$"s for soil layers lower than 83 cm were not tuned -- using the default value generated by CLM at the US-SRM site. Therefore, at each site, "without HR" and "with HR" simulations used identical parameter "$B$" tuned for that site. We then examined whether for these eight ecologically diverse sites, CLM4.5 with and/or without HR were able to reproduce basic patterns observed at the sites in ET, soil moisture with depth, and Bowen ratio.

## 2.5 Field observations

ET, sensible heat flux, and soil moisture data at the US-SRM and US-Wrc sites were obtained from Ameriflux databases for the sites. Data for these variables at the six Southern California gradient sites were obtained from the Goulden lab (http://www.ess.uci.edu/~california/). Observed soil moisture was available for multiple soil layers with the maximum depth of 200 cm and 100 cm at the US-Wrc and US-SRM sites, respectively. Soil moisture data at CA sites were processed as described in the Notes S2. Briefly, each CA site had four CS-616 water content reflectometers (three reflectometers at US-SCd), each sensing 0-30 cm depth. All CA sites except US-SCd also had five CS-229 thermal dissipation probes sensing water potential at five depths (to 200 cm). Data from both soil moisture sensor types at the Southern California sites were used conservatively. Though sensor output suggesting nighttime increases in soil moisture followed by daytime decreases is often used in the literature as a signature of HR, we only recognized such oscillations from the 0-30 cm CS-616 probes as signatures of HR if they were clearly stronger than a putative temperature-induced oscillation in surrounding portions of the signal trace (e.g. Fig. S1a) and if wavelet transform analysis of the CS-229 probe data corroborated the HR (Notes S2, Fig. S1b, c).



## 3 Results

### 3.1 Soil moisture observations and simulations

Observed soil moisture (grey lines) and CLM4.5 model simulations with (blue lines) and without (red lines) HR are plotted in Fig. 1 for selected years, for the top 0-30cm increment and also at multiple depths where such data are available. As noted

above, CS-229 thermal dissipation probes were installed from 0 to 200 cm depth at five of the six California sites, but are known only to provide reliable information down to approximately -2.5 MPa; sensor output thus flatlined for lower water potentials during drought. We therefore chose only to include 0-30 cm CS-616 probe data in Fig. 1, with panels ordered from West (US-SCs, Coastal Sage) to East (US-SCd, Sonoran Desert) down the panels. However, modeled output by depth increment at the five instrumented US-SC Southern California sites is plotted in Figs S2-6 along with temperature-corrected

data from the CS-229 probes.

Modeled soil moisture content generally follows the magnitude and dynamics in observational data (Fig. 1), except at depth at US-Wrc. At that site, we set "$B$" – the only parameter in the soil water retention curve in the models based on the soil texture information from the biological data file at the US-Wrc Ameriflux ftp website ftp://cdiac.ornl.gov/pub/ameriflux/data/Level1/ (sandy loam and loamy sand) with the maximum value being 6.65 (Table 3).

However, Shaw et al. (2004) (and http://ameriflux.ornl.gov/fullsiteinfo.php?sid=98) report that in some locations soil at depth can approach silt to clay loam for which the range of "$B$" is 8.5 +/- 3.4 (clay loam, Clapp and Hornberger, 1978). Using a higher B value in the simulations would have reduced the difference between the simulated and observed soil moisture at depth at the US-Wrc site.

At US-SRM (Fig. 1), modeled soil moisture at depth (≥ 49 cm) was more dynamic in CLM4.5+HR (blue line) than in

CLM4.5 (red line). The dynamism is also clearly seen in the observed soil moisture data (grey lines), in both the US-SRM 60-70 and 90-100 cm depths. In CLM4.5+HR, this dynamism is caused by downward HR (hydraulic descent) when root systems redistribute rain from surface to deep soils faster than it could be delivered by percolation alone (Ryel et al., 2003). In Figs S2-6, similar measured dynamism at depth is also detected by CS-229 probes for large rain events at the five instrumented Southern California gradient sites.

As discussed in the Notes S2, using wavelet analysis of site measurement data, we found clear evidence of upward HR at the most moist Southern California site US-SCf (Oak Pine Forest), and spotty evidence at US-SCw (Pinyon Juniper Woodland) and US-SCc (Desert Chaparral) sites (Fig. S1). We did not find clear phase-based evidence of upward HR at US-SCg (Grassland) or US-SCs (Coastal Sage) sites, and temperature oscillations at the US-SCd (Sonoran Desert) site were very large, precluding easy identification of periods of upward HR. Still, the CLM4.5+HR results suggested that HR could

occur at the Southern California sites given the rooting distribution of plants and the seasonal drought, but its hydrological effect on landscape-level eddy flux was predicted to be far lower where plant biomass was small (e.g. US-SCd). This combination of factors (drought, rooting depth, density of vegetation) influenced the simulated magnitude of soil moisture fluctuations, and we plot them with the sensor data in Fig. 2 and Fig. S7. The noticeable discrepancy between modeled and



measured rainy season soil moisture at the US-SCd site (indicated with grey rectangular box in Fig. 1) are most likely caused by the incomplete precipitation record (Notes S3).

Overall, Fig. 1 and the corresponding Root Mean Squared Error (RMSE) illustrate clear improvement of the match between modeled and observed soil moisture at the US-SRM site by incorporating HR into CLM4.5 (Table 6, Fig. S8). At the Southern California sites, the match is improved at the US-SCs, g, and f sites during dry periods (Table 6, Fig. S8); inclusion of HR makes little difference at the US-SCw, c, and d sites (Table 6, Fig. S8). Improvement of simulated soil moisture at shallow layers (e.g. 0-30 cm, 17-29 cm) was observed at the US-Wrc site during dry periods by incorporating HR (Table 6, Fig. S8), but at depth, the modeling challenges associated with the Clapp and Hornberger (1978) B factor (described above) precluded detection of any change in RMSE with inclusion of HR in CLM4.5. The reduced model performance in soil moisture modeling at depth by including HR at site like US-Wrc is an unnegligible challenge in HR modeling.

### 3.1.1 HR flux simulations

To evaluate the simulation of the HR flux, the modeling results were compared to both direct measurement of HR flux itself and measurement of soil moisture dynamics from which HR could be inferred. These include: (a) observed downward sap flow at the US-SRM site, (b) observed diel fluctuations of soil moisture for depth of 0-30 cm during dry periods at all eight sites, (c) the vertical change of the magnitude of observed diel fluctuations of soil moisture at the US-Wrc and US-SRM sites, and (d) the seasonal pattern of HR's influences on soil moisture at the US-Wrc site.

At the US-SRM site, Scott et al. (2008) monitored sap flow and estimated hydraulic descent during days 31-109 in 2004 to be 12-38 mm $H_2O$ $d^{-1}$; the CLM4.5+HR estimate for the same period was 35mm $H_2O$ $d^{-1}$, within the scope provided by Scott et al. (2008). CLM4.5+HR could largely capture the amplitude of the HR-induced diel fluctuations of soil moisture for depth of 0-30 cm at US-Wrc, US-SRM, US-SCs, US-SCg, and US-SCf sites during drought (Fig. 2; Fig. S7). The simulated amplitude of diel fluctuation during the dry periods decreased from shallower to deeper layers at all eight sites. For example, the simulated amplitude decreased from 0.002 at depth of 2-5 cm to essentially zero at depth of 17-29 cm at the US-SRM site, and the decrease of amplitude with depth is quantitatively consistent with observations at the US-Wrc and US-SRM sites (results now shown). At the US-Wrc site, the maximum depth of HR-induced soil moisture increases during dry seasons (mainly limited to the upper 60 cm) and the seasonal pattern of HR's influences on soil moisture could also be correctly reproduced by the CLM4.5+HR (as shown in detail in 'Soil moisture simulations with and without HR' and discussion sections). As discussed in the 'HR model parameterization' section, we used soil suction to roughly control the magnitude of HR at the three drier CA sites, where the diel fluctuation of soil moisture was clearly influenced by temperature. These comparisons indicate that the HR flux is properly simulated in the present study.





### 3.1.2 Soil moisture simulations with and without HR

Differences between CLM4.5 and CLM4.5+HR in modeled volumetric soil moisture are plotted in Fig. 3 and Fig. S9 for all sites. Inclusion of HR in CLM4.5+HR increased summertime soil moisture by several percent (above the zero line) in the six Southern California US-SC sites (0-30 cm depths), US-Wrc and US-SRM (0-49 cm depths) sites (Fig. 3). In the US-Wrc

model profile, these periods of increased shallow soil moisture clearly coincide with decreased soil moisture at depth (49-380 cm depth), consistent with hydraulic lift. In the US-SRM (Fig. 3) and Southern California US-SC site model profiles (Fig. S9), the patterns of soil moisture with depth are more complex, with central layers being sources or sinks of water depending on time of year and year itself. During rainy winter seasons at the six Southern California US-SC sites, CLM4.5+HR produced periods of reduced soil moisture in shallow 0-30 cm layers in all years at US-SCs (Coastal Sage) and

US-SCg (Grassland) sites, consistent with hydraulic descent (Fig. 3). Similar patterns are most clear only during the wettest winter 2011 for US-SCd (Sonoran Desert), SCc (Desert Chaparral), SCw (Pinyon Juniper), and SCf (Oak Pine) sites.

Pulling together averaged model output from all years, for 0-250 cm depths at each site, Fig. 4 illustrates the complex patterns in the change in volumetric soil water content driven by inclusion of Ryel et al.'s (2002) HR model in the CLM4.5 modeling framework, over the annual cycle. Blue indicates an increase of up to 1% volumetric soil moisture in the

15 CLM4.5+HR vs. the CLM4.5 model output. Yellow indicates a decrease of up to 4% volumetric soil moisture in the CLM4.5+HR vs. CLM4.5 model output. Isolines are labeled with % soil moisture change in the CLM4.5+HR vs. CLM4.5 cases. (Contours are generated from soil moisture increases or decreases in each CLM4.5-defined layer node; node depths increase exponentially downward). At the wettest site US-Wrc, modeled upward HR (hydraulic lift) is mainly concentrated during the dry season July-September (~days 180-240), followed by HR downward (hydraulic descent) during October.

Effects of HR on modeled soil water content persist for a longer time during the year at the other seven sites. At the US-SRM site, hydraulic lift was most evident in May and June (just before the North American monsoon season July-September); hydraulic descent could be found almost throughout the year, and the most significant hydraulic descent occurred during the monsoon season. Among the six Southern California sites, a gradient in the depth and temporal extent of HR on modeled soil moisture was clear. The largest increases (and decreases) in soil moisture occurred at the most moist

(but still seasonally dry) US-SCf (Oak Pine) site with deciduous oak trees, followed by US-SCg (Grassland) and US-SCs (Coastal Sage), and US-SCw (Pinyon Juniper). At the much drier US-SCc (Desert Chaparral) and US-SCd (Sonoran Desert) sites with sparse vegetation, the temporal spread and depth of influence of HR were far more limited. Still, hydraulic descent occurred during at least a small portion of December (between days 330-365) at all Southern California US-SC sites.

Table 5 shows the average modeled hydraulic lift (in mm d$^{-1}$) during dry periods, for all sites; highest values were found

at the two forested sites with highest annual precipitation (0.71 and 0.60 mm H$_2$O d$^{-1}$ for US-SCf and US-Wrc sites, respectively). Modeled hydraulic lift is comparatively small at the US-SRM (0.19 mm H$_2$O d$^{-1}$) and the three drier Southern California sites (US-SCw, US-SCc, and US-SCd: 0.10-0.22 mm H$_2$O d$^{-1}$).



## 3.2 Evapotranspiration observations and simulations

Fig. 5, documents the model performance in simulating ET at the daily time scale, at all eight study sites, over multiple years. Fig. 5 shows that CLM4.5+HR can simulate ET well at the US-Wrc and US-SRM sites, but tends to underestimate ET during the high ET periods at the six Southern California sites. An increase in modeled ET associated with HR during

drought can be identified (to varying degrees) at all eight sites. Fig. 5 and the corresponding RMSE (Table 6) illustrate that including HR leads to improvement in ET simulation at the US-SRM and US-SCf sites during dry periods and year round, and also improvement at the US-SCs and US-Wrc sites during dry periods. At other sites, the corresponding ET simulations from CLM4.5+HR and CLM4.5 are very similar.

Fig. 6 shows the average diel cycles of ET and its components (hourly) during dry and wet periods, at the eight sites

(Notes S4). From Fig. 6, CLM4.5+HR tends to underestimate the observed mid-day ET peak around noon at the US-Wrc, US-SRM, US-SCf, and US-SCw sites, but reproduced observations fairly well at the US-SCc and US-SCd sites. HR-induced increase in simulated mid-day transpiration and subsequent increase of ET can be identified during the dry periods at all eight sites, though it is very weak at US-SCc and US-SCd. Compared to dry periods, HR-induced changes in simulated ET were relatively limited during wet periods at all eight study sites. At the US-SRM site, a decrease of ground evaporation

and increase of transpiration were both evident during wet periods, caused by significant hydraulic descent at this site (Fig. 1, 4). The soil water in shallow layers that would otherwise be evaporated was redistributed to deep layers during and after rain events in the monsoon season (July-September), and was subsequently taken up from deeper layers by plants during transpiration.

Table 5 shows the HR-induced increase in ET (mm $H_2O$ $d^{-1}$), estimated as the difference in ET between simulations with

and without HR. The contribution of HR to ET (unit: %) refers to this difference normalized by the ET from CLM4.5+HR. The HR-induced ET increase (0.47 mm $H_2O$ $d^{-1}$) is largest at the US-SCf site, and corresponding ET increase is comparatively small at the US-SRM (0.18 mm $H_2O$ $d^{-1}$) and the three drier Southern California sites (US-SCw, US-SCc, and US-SCd: 0.06-0.13 mm $H_2O$ $d^{-1}$) (Table 5).

## 3.3 HR-induced Bowen ratio change

The partitioning of surface energy between latent and sensible heat fluxes, often characterized using the Bowen Ratio (the ratio of sensible heat to latent heat flux), drives the dynamics of boundary layer growth and subsequently the triggering mechanisms of convective precipitation (Siqueria et al., 2009). The influence of HR on Bowen ratio is therefore important for understanding the broader impact of HR beyond the land surface. Including HR improves the model performance in reproducing the Bowen ratio (Fig. 7, Table 6), especially during dry periods, at all sites except the two driest Southern

California sites (US-SCc and US-SCd). This indicates that the ET or soil moisture comparison alone does not capture the full benefit of including HR in the model. Instead, HR's impact on ET and soil moisture influences surface temperature and therefore sensible heat flux. The Bowen ratio synthesizes these effects of HR. The better agreement between model and



observation in Bowen ratio than in ET may be related to the challenge of the eddy covariance flux measurement. Since ET (latent heat flux) and sensible heat flux are both derived from the same eddy covariance measurement, potential errors in quantifying the eddy covariance (which are not uncommon as reflected by the energy closure challenge facing many flux tower measurements) are likely to have a much smaller impact on the Bowen ratio estimate than on the magnitude of latent

heat flux or sensible hear flux alone.

Combining the modeling results for daily ET into Fig. 8, a larger pattern emerges from the cross-site comparison. Each site is color-coded differently, and HR-induced increases in ET are plotted against shallow soil moisture (0-30 cm, commonly measured at field sites beyond those studied here). At low soil moisture, the driest Southern California gradient sites have little water to redistribute and very sparse vegetation to carry out HR. At high soil moisture, little driving gradient

exists to support HR. By including all sites in Fig. 8, it is clear that HR-induced increases in ET are maximal at sites with mid-range soil moistures.

### 3.4 Sensitivity to HR model parameters

The sensitivity of modeled hydraulic lift, hydraulic descent, and contribution of HR to ET (defined in Table 5) to parameters $C_{RT}$, $\varphi_{50}$, and $b$ in the "Ryel et al. 2002" equation was tested for four sites (US-Wrc, US-SRM, US-SCs and US-SCw). Both

hydraulic lift and hydraulic descent were nearly insensitive to variation in $b$ (ranging from 0.22 to 4.22) (Figs S10, S11). Variation of approximately an order of magnitude in $C_{RT}$ (from 0.1 to 1.5 cm MPa$^{-1}$ h$^{-1}$) and $\varphi_{50}$ (from -0.5 to -4.0 MPa) resulted in less than a doubling of the magnitude of hydraulic lift, even at the height of HR (Fig. S10). However, hydraulic descent was notably more sensitive; increasing $C_{RT}$ from 0.1 to 1.5 cm MPa$^{-1}$ h$^{-1}$ resulted in nearly an order of magnitude increase in maximum  hydraulic descent at US-Wrc (from ~0.1 to ~1 mm d$^{-1}$), and a tripling of hydraulic descent at the other

sites (Fig. S11). A change in $\varphi_{50}$ from -4.0 MPa to -0.5 MPa led to at most a tripling of hydraulic descent at all sites. Similarly, the modeled contribution of HR to ET was sensitive to $C_{RT}$ and $\varphi_{50}$ and insensitive to $b$ (Fig. 9).

### 4 Discussion

Measurement and modeling both demonstrate that the timing, magnitude, and direction (upward or downward) of HR vary across ecosystems (Figs 1, 4), and incorporation of HR into CLM4.5 improved model-measurement match particularly

during dry seasons (Table 6). The hydrological impact of HR is substantial in ecosystems that have a pronounced dry season but are not overall so dry that sparse vegetation and very low soil moisture limit HR (Figs 4, 7, 8). The lack of HR representation in the current generation of land surface or earth system models thus should be considered a source of error when modeling seasonally-dry ecosystems with deep-rooted plant species.



## 4.1 HR-induced soil moisture change

Several of the AmeriFlux sites investigated here have hosted previous field investigations of impacts of HR on soil moisture. CLM4.5+HR was able to capture patterns published from those empirical data, and added to those data a more comprehensive view of the seasonal dynamics in the systems (Fig. 4). For example, at US-Wrc (a ~450-year old stand of Douglas Fir), the CLM4.5+HR results indicated that HR-induced soil moisture increases during dry seasons were mainly limited to the upper 60 cm of soil (Fig. 4), which is consistent with field measurements (soil moisture and soil water potential) in a ~20-year-old and a ~450-yr-old Douglas Fir stand in the Pacific Northwest (Brooks et al., 2002; Brooks et al., 2006; Meinzer et al., 2004). The US-Wrc panel in Fig. 4 also shows that as soil drying progressed, more water was redistributed to depth of 20-60 cm from lower layers in late summer than in early summer. (It is worth noting that the CLM4.5+HR model does not include the temperature fluctuation-driven vapor transport within soil shown by Warren et al. (2011) to occur at the site.)

US-Wrc is the site with the highest annual rainfall (> 2m per year) among those modeled (Table 1), and the effects of HR are constrained to the mid-year dry season and dominated by hydraulic lift (Fig. 4). Hydraulic descent is limited, averaging 5.0 mm H2O yr$^{-1}$ during 1999-2012, perhaps because soil moisture is higher with depth, limiting the driving gradient for hydraulic descent. In contrast, hydraulic lift and hydraulic descent are active nearly year-round at five of the other seven AmeriFlux sites (Fig. 4). At the two driest sites US-SCc and US-SCd, due to the scarcity of water that can be moved and the sparse vegetation, the HR-associated dynamics in water content are relatively subdued (Fig. 4). At the US-SCf site, Kitajima et al. (2013) simulated hydraulic lift from 2007 to 2011 using the HYDRUS-1D model on a daily scale (without simulating the diel fluctuation of soil moisture), and the simulated hydraulic lift averaged ~ 28 mm per month in July and August, which is close to the 24.7 mm per month from CLM4.5+HR. The annual hydraulic lift was ~112 mm in Kitajima et al. (2013), and was 121 mm in CLM4.5+HR. However, the two modeling approaches are quite different. Kitajima et al. (2013) attributed the source of hydraulic lift to deep moisture in the weathered bedrock, and did not account for the hydraulic redistribution within the soil layers. In contrast, CLM4.5+HR included HR among the soil layers but not the hydraulic lift from deep bedrock. Hydraulic descent occurring after rain was not included in Kitajima et al. (2013), but featured prominently at year end in the output from CLM4.5+HR (Fig. 4, panel US-SCf, right-hand side). The hydraulic lift from deep bedrock is also a possible reason for the reduced model performance in soil moisture modeling at depth for site like US-Wrc.

Though sap flow indicated little hydraulic lift during 2004-2005 (Scott et al., 2008), CLM4.5+HR-simulated hydraulic lift is significant during dry periods at the US-SRM site (Fig. 4), and diel fluctuations of soil moisture indicative of HR were observed during soil drydown (Fig. 2). Scott et al. (2008) calculated hydraulic descent using the downward flow in taproots, and calculated hydraulic lift using lateral root flow moving away from the tree base. Flow was more concentrated and more easily measured in the taproot than in lateral roots, which was considered as the reason why the monitored hydraulic descent was far more detectable than hydraulic lift.



## 4.2 HR-induced evapotranspiration change

The influence of HR on transpiration and/or ET has been estimated in many studies, including at sites modeled here. At the US-Wrc site, Brooks et al. (2002) used diel fluctuations in soil moisture, and total soil water use, to calculate that HR contributed about 28 % to the total daily water use from the upper 2 m of the soil in a 20-year-old Douglas-fir stand during dry August, similar to the 32 % estimated here (Table 5). At the US-SRM site and seasonal scale, Scott et al. (2008) reported that the hydraulic descent during the dormant season (DOY 31-109) represented 15-49 % of the estimated growing season (DOY 110-335) transpiration in 2004; the corresponding simulated value during the same period in the present study is 36 %. ET was notably underestimated at the US-SCf site by both CLM4.5+HR and HYDRUS-1D (Kitajima et al., 2013). The lack of hydraulic lift from bedrock in this study, and the lack of HR within soil layers in Kitajima et al. (2013), might be reasons for this underestimation.

## 4.3 Sources of uncertainty

Results in this study are subject to uncertainties from a number of sources. As noted in the methods, data essential for the CLM4.5 and HR models were drawn from each site when available, but otherwise were drawn from large datasets commonly used in large-scale models (Table 2). Also as noted in the methods and Notes S2, soil moisture measurements were challenging at the Southern California sites because large temperature gradients developed along CS616 probes, soils dried outside the range of CS-229 probes, and there appeared to be a thermal gradient between reference thermistor and sensor connection points in measurement junction boxes aboveground. More subtle and interesting sources of uncertainty also likely influenced the model-measurement match. For example, strong inter-annual variation of precipitation, fire, and recovery from fire caused rather abrupt changes of PFT coverage and LAI at the US-SCs site. The US-SCg site is undergoing restoration to a native grassland community, and a large community of ephemeral annuals comes up following winter or summer rains at the US-SCc site. These variations are difficult to capture by satellite remote sensing data but undoubtedly affected soil moisture and ET in interesting ways. Without detailed ground-observational data to quantify them, simulations in this study used a climatological LAI seasonal cycle.

Another potentially important source of uncertainty is the parameters $C_{RT}$, $b$, and $\varphi_{50}$ in the HR model. Quantifying these parameters remains a major challenge. Results from our sensitivity experiments show that CLM4.5+HR output is relatively insensitive to variation in the parameter b, so of the three parameters, giving b a default value is least problematic. As shown in Ryel et al. (2002), maximum conductance $C_{RT}$ can be determined from site-specific data (soil moisture, soil water potential, and root distribution). But in the absence of such data, an approach might be developed based on the idea that in any ecosystem, there must be sufficient maximum soil-whole plant conductance ($C_{RT}$) to support the annual maximum observed LAI when soil is saturated (Wullschleger et al. 1998). Determining a reasonable way to estimate $\varphi_{50}$ may require the most effort. Field measurements combined with modeling may be necessary to enable setting the value of $\varphi_{50}$ and to ground-truth a relationship between $C_{RT}$ and annual maximum LAI, ideally across a range of ecosystem types, vegetation





densities, soil textures, and/or other site-specific properties that are already input variables for earth system models. In addition, the effects of root architecture, deep water uptake (Markewitz et al., 2010), and HR representation models (Amenu and Kumar, 2008; Quijano and Kumar, 2015) on the quantitative estimates of HR, temperature fluctuation-driven vapor transport within soil, and how to distinguish hydraulic descent from macropore flow (Fu et al., 2012, 2014), all need further investigations.

## 5 Main Findings

The key findings in this study include,

- Simulated hydraulic lift was largest at the two forested sites with highest annual rainfall (0.60 US-Wrc and 0.71 mm $H_2O$ $d^{-1}$ US-SCf), and smallest at US-SRM and the three driest Southern California sites (0.10 US-SCc to 0.22 mm $H_2O$ $d^{-1}$ US-SCw).

- Hydraulic descent was a dominant hydrologic feature during wet seasons at semi-arid US-SRM (Fig. 1, 4) and four (moister) of the six Southern California sites (Fig. 4, Figs S2-6) with annual precipitation $\leq \sim 500$ mm (Table 1), contributing to significant dynamism in soil moisture at depth.

- HR caused modeled ET to increase, particularly during dry periods; values for the increase ranged from 0.06, 0.10, and 0.13 mm $H_2O$ $d^{-1}$ at the driest sites (US-SCc, US-SCd, and US-SCw, respectively) to 0.18, 0.26, 0.29, 0.35, and 0.47 mm $H_2O$ $d^{-1}$ at the wetter sites (US-SRM, US-SCs, US-Wrc, US-SCg, and US-SCf, respectively).

- Measurement and modeling both demonstrate that the timing, magnitude, and direction (upward or downward) of HR vary across ecosystems, and incorporation of HR into CLM4.5 improved model-measurement match for Bowen ratio, evapotranspiration, and soil moisture (e.g. shallow layers), particularly during dry seasons.

- Modeling and measurements indicate that HR has hydrological impact (on evapotranspiration, Bowen ratio, and soil moisture) in ecosystems that have a pronounced dry season but are not overall so dry that sparse vegetation and very low soil moisture limit HR.

- CLM4.5+HR output was relatively insensitive to variation in the parameter b in the Ryel et al. 2002 equation, but was somewhat sensitive to variation in $C_{RT}$ and $\varphi_{50}$. Variation of approximately an order of magnitude in $C_{RT}$ and $\varphi_{50}$ resulted in less than a doubling of the magnitude of hydraulic lift, but hydraulic descent was more sensitive.

Previous modeling studies either focus on model-data comparison at one site or conduct large scale simulations with few concrete data to compare against, making it very difficult to answer the fundamental question: When and where must HR be included to appropriately model hydrologic characteristics of diverse ecosystems? HR has been confirmed in many ecosystems where plant root systems span soil water potential gradients (Neumann and Cardon, 2012; Prieto et al., 2012; Sardans and Peñuelas, 2014). For this reason, one might argue that HR should be included for all ecosystems. However, our comparative study using combined empirical data and modeling helps hone the answer by including a large number of



AmeriFlux sites that differ in vegetation, soil, and climate regimes. The summary suggestions are (a) hydrological modeling will not be clearly influenced if not including HR for overall drier sites that have little water to redistribute and sparse vegetation to carry out HR and overall wetter sites / periods that are likely to develop little driving gradient to support HR, while HR should be included for the seasonally dry ecosystems with mid-range annual rainfall and soil moisture, and (b) quantifying parameters in the HR model is a key if including HR in hydrological modeling.

## Acknowledgements

This research was supported by the Office of Science (BER), U.S. Department of Energy (DE-SC0008182 ER65389). Funding for the eight Ameriflux sites was also provided by the U.S. Department of Energy's Office of Science.

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





**Supporting Information**

**Fig. S1** Detection of HR using wavelet analysis. (a) CS-229 probe traces from the Oak Pine Forest (US-SCf) site, during late June and early July, 2011. (b) Wavelet analysis, comparing the period and phase of oscillations in 5, 10, and 100 cm probes with the 200 cm probe. (c) Subset of wavelet analysis during mid-April, 2011, at the Desert Chaparral (US-SCc) site, comparing period and phase of oscillations in 10 & 50 cm probes with the 100 cm probe. In (b) and (c), the thick black contour designates the 95% confidence level against AR(1) red noise, and the cone of influence where edge effects affect interpretation is shown as a lighter shade. The period unit is day. Large power in (b) and (c) implies clear fluctuation with a certain periodicity in a certain time frame. (d) Representative relationships between CS-616 probe output (integrating moisture from 0-30 cm), and a 5cm (red dots) and 25 cm (blue dots) CS-229 probe.

**Fig. S2** Observed and simulated soil moisture from 5 to 200cm depth, over multiple years, at the US-SCs Coastal Sage site. Depths of observed and simulated soil moisture are shown at upper right in each panel. Grey lines included in the 5cm, 10cm, and 25 cm layers are data from the four CS-616 soil moisture sensors integrating information from 0-30 cm depth. Blue and red lines are simulated soil moisture from CLM4.5+HR and CLM4.5, at the depths indicated. Each depth panel also contains one colored line (magenta, pink, orange, green, cyan, light blue, for 5, 10, 25, 50, 100, and 200 cm CS-229 probes, respectively).

**Fig. S3** Observed and simulated soil moisture from 5 to 200cm depth, over multiple years, at the US-SCg Grassland site. Lines as described for Fig. S2.

**Fig. S4** Observed and simulated soil moisture from 5 to 200cm depth, over multiple years, at the US-SCf Oak Pine Forest (James Reserve) site. Lines as described for Fig. S2.

**Fig. S5** Observed and simulated soil moisture from 5 to 200cm depth, over multiple years, at the US-SCw Pinyon Juniper Woodland site. Lines as described for Fig. S2.

**Fig. S6** Observed and simulated soil moisture from 5 to 200cm depth, over multiple years, at the US-SCc Desert Chaparral site. Lines as described for Fig. S2.

**Fig. S7** Observed and simulated diel fluctuations of soil moisture for depth of 0-30 cm during dry periods. One specific observed (simulated) soil moisture fluctuation is firstly cut by the straight line between the minimum soil moisture of two adjacent days, then, the subsequent diel fluctuations are averaged over the entire dry period (defined Table 5). SW1-4 for Southern California sites are from four CS-616 water content reflectometers (each sensing 0-30 cm depth). SW1 is not available at the US-SCd site.

**Fig. S8** Root Mean Squared Error quantifying the match between modeled and observed soil moisture shown in Fig. 1. For the Southern California sites, data for each individual CS-616 probe (four per site) are shown.





**Fig. S9** Soil moisture difference between simulations with HR and without HR at each southern California site (US-SCs,g,f,w,c).

**Fig. S10** Sensitivity of simulated upward HR (hydraulic lift) to selected parameters in Ryel et al.'s 2002 equation describing HR. Parameters are shown in Tables 3 and 4 except those shown in each sub-figure. Results shown here are the averaged values for Julian days over the entire simulation period. Interruptions are periods when hydraulic descent occurs.

**Fig. S11** Sensitivity of simulated downward HR (hydraulic descent) to selected parameters in Ryel et al.'s 2002 equation describing HR. Parameters as in Fig. S10.

**Notes S1** Parameterization of CLM4.5 when site-specific data were not available from AmeriFlux sites.

**Notes S2** Site-specific data processing of thermal dissipation probe data at Southern California gradient sites.

**Notes S3** Discrepancies between modeled and measured rainy season soil moisture at the Southern California gradient sites.

**Notes S4** Partitioning of evapotranspiration into transpiration and ground and canopy evaporation.



**Table 1.** Study site information.

| Site | Location | Elevation (m) | Climate | Vegetation | Annual precipitation (mm) | Average temperature (ºC) | Atmospheric forcing data |
|---|---|---|---|---|---|---|---|
| Wind River Crane Site (US-Wrc) | 45.8205ºN, 121.9519ºW, WA | 371 | Mediterranean (Csb) | Douglas-fir/western hemlock | 2223[1#] | 8.7[1#] | 1999-2012 |
| Santa Rita Mesquite (US-SRM) | 31.8214ºN, 110.8661ºW, AZ | 1116 | Cold semi-arid (BSk) | Mesquite tree, Grass | 377[2#] | 19.6[3#] | 2004-2012 |
| Southern California Climate Gradient - Coastal Sage (US-SCs) | 33.7342ºN, 117.6961ºW, CA | 475 | Mediterranean (Csa) | Coastal Sage | 288[4#] | 16.2[4#] | 2007-2012 |
| Grassland (US-SCg) | 33.7364ºN, 117.6947ºW, CA | 470 | Mediterranean (Csa) | Grass | 281[4#] | 16.6[4#] | 2007-2012 |
| Oak Pine Forest (US-SCf) | 33.8080º N, 116.7717ºW, CA | 1710 | Mediterranean (Csa) | Oak/pine forest | 526[4#] | 13.3[4#] | 2007-2012 |
| Pinyon Juniper Woodland (US-SCw) | 33.6047º N, 116.4527ºW, CA | 1280 | Mediterranean (Csa) | Pinyon, juniper | 100[4#] | 16.5[4#] | 2007-2012 |
| Desert Chaparral (US-SCc) | 33.6094ºN, 116.4505ºW, CA | 1300 | Mediterranean (Csa) | Desert shrubland | 153[4#] | 16.3[4#] | 2007-2012 |
| Sonoran Desert (US-SCd) | 33.6518ºN, 116.3725ºW,CA | 275 | Mediterranean (Csa) | Desert perennials and annuals | 123[5#] | 23.8[5#] | 2007-2011 |

Notes: 1#: 1978-1998, statistic is based on a NOAA station located 5 km north of the US-Wrc tower. 2#: 1937-2007, from Scott et al. (2009). 3#: 2004-2012. 4#: 2007-2012. 5#: 2007-2011.





**Table 2.** Sources of data for model inputs.

| Site | Atmospheric forcing data | Land coverage | LAI | Canopy height | Soil texture | Soil organic matter |
|------|--------------------------|---------------|-----|---------------|--------------|---------------------|
| US-Wrc | AmeriFlux tower data | Google Earth map; Table 2 in Shaw et al. (2004) (overstory trees: 24%; vine maple: 36%; salal and oregon grape: 40%) | Shaw et al. (2004); AmeriFlux biological data file | Table 3 in Shaw et al. (2004) (mean overstory tree height: 19.2 m) | Fig. 4 in Warren et al. (2005). Sandy loam, with loamy sand at some depths. | Table 1 in Shaw et al. (2004); AmeriFlux biological data file |
| US-SRM | AmeriFlux tower data | Dr. Russell Scott from USDA-ARS (bare ground: 40%; mesquite canopy: 35%; grass: 25%) | Dr. Russell Scott from USDA-ARS | Potts et al. (2008) (Tree height: 0.25-5) | AmeriFlux biological data file. Mixed sandy loam and loamy sand. | AmeriFlux biological data file |
| US-SCs | UCI Goulden Lab | Estimation based on site picture (bare ground: 10%; coastal sage: 90%) | NCAR database | NCAR database | UCI Goulden Lab Shallow sand, deep loamy sand. | NCAR database |
| US-SCg | UCI Goulden Lab | Estimation based on site picture (bare ground: 10%; grass: 90%) | NCAR database | NCAR database | UCI Goulden Lab Shallow sand, deep loamy sand. | NCAR database |
| US-SCf | UCI Goulden Lab | Table 3 in Anderson and Goulden (2011) (Doak) | Table 2 in Fellows and Goulden (2013) | NCAR database | UCI Goulden Lab Sandy loam, with loamy sand at some depths. | NCAR database |
| US-SCw | UCI Goulden Lab | Table 3 in Anderson and Goulden (2011) (Oshrub) | NCAR database | NCAR database | UCI Goulden Lab Estimated as sand (sand: 90%; clay: 7.5%) | NCAR database |
| US-SCc | UCI Goulden Lab | Google Earth map (bare ground: 78%; chaparral: 22%) | UCI Goulden Lab | NCAR database | UCI Goulden Lab Estimated as sand (sand: 90%; clay: 7.5%) | NCAR database |
| US-SCd | UCI Goulden Lab | Table 3 in Anderson and Goulden (2011) (LowDes) | NCAR database | NCAR database | UCI Goulden Lab Estimated as sand (sand: 99%; clay: 0.5%) | NCAR database |



**Table 3.** Clapp and Hornberger "*B*" used in this study.

| Layers | Depth at layer interface (m) | US-Wrc *B* | Soil texture1# | US-SRM *B* | Soil texture1# | US-SCs *B* | Soil texture1# | US-SCg *B* | Soil texture1# | US-SCf *B* | Soil texture1# | US-SCw *B* | Soil texture2# | US-SCc *B* | Soil texture2# | US-SCd *B* | Soil texture2# |
|---|---|---|---|---|---|---|---|---|---|---|---|---|---|---|---|---|---|
| 1 | 0.0175 | 3.96 | SL | 3.15 | LS | 5.07 | S | 4.46 | S | 3.15 | SL | 4.09 | S | 4.09 | S | 2.27 | S |
| 2 | 0.0451 | 4.31 | SL | 3.15 | LS | 5.09 | S | 4.49 | S | 3.26 | SL | 4.09 | S | 4.09 | S | 2.27 | S |
| 3 | 0.0906 | 4.46 | SL | 3.15 | LS | 5.13 | S | 4.53 | S | 3.39 | SL | 4.10 | S | 4.10 | S | 2.27 | S |
| 4 | 0.1655 | 4.52 | SL | 3.16 | LS | 5.30 | LS | 4.65 | S | 3.18 | SL | 4.11 | S | 4.11 | S | 2.27 | S |
| 5 | 0.2891 | 4.39 | SL | 3.41 | LS | 4.83 | LS | 4.27 | LS | 3.34 | SL | 4.11 | S | 4.11 | S | 2.27 | S |
| 6 | 0.4929 | 4.31 | SL | 3.66 | LS | 4.63 | LS | 4.19 | LS | 3.27 | SL | 4.11 | S | 4.11 | S | 2.27 | S |
| 7 | 0.8289 | 4.00 | SL | 3.91 | LS | 3.94 | LS | 4.33 | LS | 3.27 | LS | 4.11 | S | 4.11 | S | 2.27 | S |
| 8 | 1.3828 | 5.85 | LS | 4.41 | LS | 3.51 | LS | 4.08 | LS | 3.30 | SL | 4.11 | S | 4.11 | S | 2.27 | S |
| 9 | 2.2961 | 6.65 | SL | 4.40 | LS | 3.15 | LS | 3.90 | LS | 3.50 | SL | 4.10 | S | 4.10 | S | 2.27 | S |
| 10 | 3.8019 | 6.65 | SL | 4.40 | LS | 3.15 | LS | 3.90 | LS | 3.50 | SL | 4.10 | S | 4.10 | S | 2.27 | S |

Note: 1#-derived from soil sample data in former studies; 2#-estimated by UCI Goulden Lab. "S" represents sand, "LS" loamy sand, and "SL" sandy loam. *B* values for sand, loamy sand, and sandy loam were 2.27-5.83, 2.91-5.85, and 3.15-6.65 in Clapp and Hornberger (1978), respectively.





**Table 4.** Parameters used in the "Ryel et al. 2002" HR model for the study sites. "$C_{RT}$" is the maximum radial soil−root conductance, "$\varphi_{50}$" is the soil water potential where conductance is reduced by 50%, "$b$" is an empirical constant.

| Site | $C_{RT}$ (cm MPa$^{-1}$ h$^{-1}$) | "$b$" | $\varphi_{50}$ (MPa) |
|---|---|---|---|
| US-Wrc | 0.1 | 3.22 | -1.0 |
| US-SRM | 1.0 | 3.22 | -1.0 |
| US-SCs | 1.0 | 3.22 | -1.0 |
| US-SCg | 0.25 | 3.22 | -1.0 |
| US-SCf | 1.0 | 3.22 | -1.0 |
| US-SCw | 0.1 | 3.22 | -1.0 |
| US-SCc | 0.05 | 3.22 | -1.0 |
| US-SCd | 0.05 | 3.22 | -1.0 |



**Table 5**. Modeled contribution of hydraulic redistribution (HR) to evapotranspiration (ET) during dry periods (mean ± s.d., columns 3-6).

| Site | Dry period (month/day) | $HL$* (mm $H_2O$ $d^{-1}$) | $ET_{without\ HR}$ (mm $H_2O$ $d^{-1}$) | $ET_{with\ HR}$ (mm $H_2O$ $d^{-1}$) | HR-induced ET increase ($ET_{with\ HR} - ET_{without\ HR}$; mm $H_2O$ $d^{-1}$) | $\dfrac{HL}{ET_{with\ HR}}$ | Contribution of HR to ET ($\dfrac{ET_{with\ HR} - ET_{without\ HR}}{ET_{with\ HR}}$; %) |
|---|---|---|---|---|---|---|---|
| US-Wrc | 6/1-9/30 | 0.60 ± 0.44 | 1.61 ± 0.82 | 1.90 ± 0.77 | 0.29 ± 0.35 | 0.32 | 0.15 |
| US-SRM | 5/1-6/30 | 0.19 ± 0.10 | 0.34 ± 0.45 | 0.52 ± 0.39 | 0.18 ± 0.20 | 0.37 | 0.34 |
| US-SCs | 4/1-9/30 | 0.41 ± 0.18 | 0.63 ± 0.50 | 0.89 ± 0.48 | 0.26 ± 0.17 | 0.46 | 0.29 |
| US-SCg | 4/1-9/30 | 0.48 ± 0.13 | 0.59 ± 0.46 | 0.94 ± 0.38 | 0.35 ± 0.19 | 0.51 | 0.37 |
| US-SCf | 4/1-9/30 | 0.71 ± 0.26 | 0.85 ± 0.50 | 1.32 ± 0.47 | 0.47 ± 0.33 | 0.53 | 0.35 |
| US-SCw | 4/1-9/30 | 0.22 ± 0.07 | 0.34 ± 0.31 | 0.47 ± 0.31 | 0.13 ± 0.10 | 0.46 | 0.29 |
| US-SCc | 4/1-9/30 | 0.10 ± 0.07 | 0.39 ± 0.42 | 0.45 ± 0.43 | 0.06 ± 0.07 | 0.21 | 0.13 |
| US-SCd | 4/1-9/30 | 0.14 ± 0.08 | 0.34 ± 0.38 | 0.44 ± 0.36 | 0.10 ± 0.08 | 0.31 | 0.22 |

\* HL represents hydraulic lift (upward HR).



**Table 6**. Root mean square error (RMSE) comparing field observations with modeled output from CLM4.5 or CLM4.5+HR.

| Site | Bowen Ratio (multi-year, dry period) | | | Evapotranspiration (multi-year, dry period) | | | Soil Moisture (0-30 cm)* (multi-year, dry period) | | | Soil Moisture (middle / deep layers) (multi-year, dry period) | | |
|---|---|---|---|---|---|---|---|---|---|---|---|---|
| | CLM4.5 | | CLM4.5+HR | CLM4.5 | | CLM4.5+HR | CLM4.5 | | CLM4.5+HR | CLM4.5 | | CLM4.5+HR |
| US-Wrc | 1.36 | >** | 0.74 | 0.77 | > | 0.69 | 7.09 | > | 5.92 | 60 cm: 6.11 | < | 60 cm: 7.67 |
| | | | | | | | | | | 100 cm: 8.81 | < | 100 cm: 9.86 |
| | | | | | | | | | | 150 cm: 11.56 | < | 150 cm: 11.95 |
| | | | | | | | | | | 200 cm: 22.67 | < | 200 cm: 23.08 |
| US-SRM | 13.30 | > | 6.84 | 0.53 | > | 0.35 | 2.35 | > | 1.15 | 60 cm: 1.36 | > | 60 cm: 0.39 |
| | | | | | | | | | | 90 cm: 1.56 | > | 90 cm: 0.47 |
| US-SCs | 6.36 | > | 4.71 | 0.47 | > | 0.42 | 4.65 | > | 3.85 | – | – | – |
| US-SCg | 2.72 | > | 1.80 | 0.53 | | 0.54 | 2.67 | > | 2.46 | – | – | – |
| US-SCf | 3.35 | > | 1.09 | 1.14 | > | 0.82 | 2.67 | > | 2.45 | – | – | – |
| US-SCw | 6.04 | > | 3.02 | 0.40 | | 0.37 | 2.30 | | 2.36 | – | – | – |
| US-SCc | 5.25 | | 5.27 | 0.42 | | 0.44 | 2.38 | | 2.29 | – | – | – |
| US-SCd | 6.02 | | 6.05 | 0.28 | | 0.30 | 1.67 | | 1.51 | – | – | – |

| Site | Bowen Ratio (multi-year, dry&wet) | | | Evapotranspiration (multi-year, dry&wet) | | | Soil Moisture (0-30 cm) (multi-year, dry&wet) | | | Soil Moisture (middle / deep layers) (multi-year, dry&wet) | | |
|---|---|---|---|---|---|---|---|---|---|---|---|---|
| | CLM4.5 | | CLM4.5+HR | CLM4.5 | | CLM4.5+HR | CLM4.5 | | CLM4.5+HR | CLM4.5 | | CLM4.5+HR |
| US-Wrc | 2.87 | | 2.94 | 0.74 | | 0.71 | 8.39 | > | 8.01 | 60 cm: 5.35 | < | 60 cm: 6.17 |
| | | | | | | | | | | 100 cm: 8.59 | < | 100 cm: 9.17 |
| | | | | | | | | | | 150 cm: 14.45 | < | 150 cm: 14.65 |
| | | | | | | | | | | 200 cm: 22.15 | < | 200 cm: 22.40 |
| US-SRM | 9.13 | > | 4.11 | 0.51 | > | 0.29 | 1.00 | | 0.88 | 60 cm: 1.79 | > | 60 cm: 1.15 |
| | | | | | | | | | | 90 cm: 2.23 | > | 90 cm: 1.12 |
| US-SCs | 5.02 | > | 4.36 | 0.49 | | 0.47 | 7.18 | | 7.34 | – | – | – |
| US-SCg | 2.01 | > | 1.29 | 0.58 | | 0.61 | 4.85 | < | 5.36 | – | – | – |
| US-SCf | 2.70 | > | 0.89 | 0.94 | > | 0.70 | 3.02 | > | 2.79 | – | – | – |
| US-SCw | 5.33 | > | 2.85 | 0.38 | | 0.36 | 3.51 | < | 3.87 | – | – | – |
| US-SCc | 3.80 | | 3.98 | 0.41 | | 0.42 | 3.60 | | 3.59 | – | – | – |
| US-SCd | 4.44 | | 4.31 | 0.27 | | 0.28 | 2.43 | > | 2.19 | – | – | – |

*Southern California observed soil moistures were calculated from the average of four (or three, for US-SCd) soil moisture probes.

**Differences between RMSE for CLM4.5 and CLM4.5+HR larger than 0.2 (for Bowen Ratio and Soil Moisture) and 0.05 (for Evapotranspiration) are indicated with ">" or "<". Smaller RMSE indicates improved model fit to data.





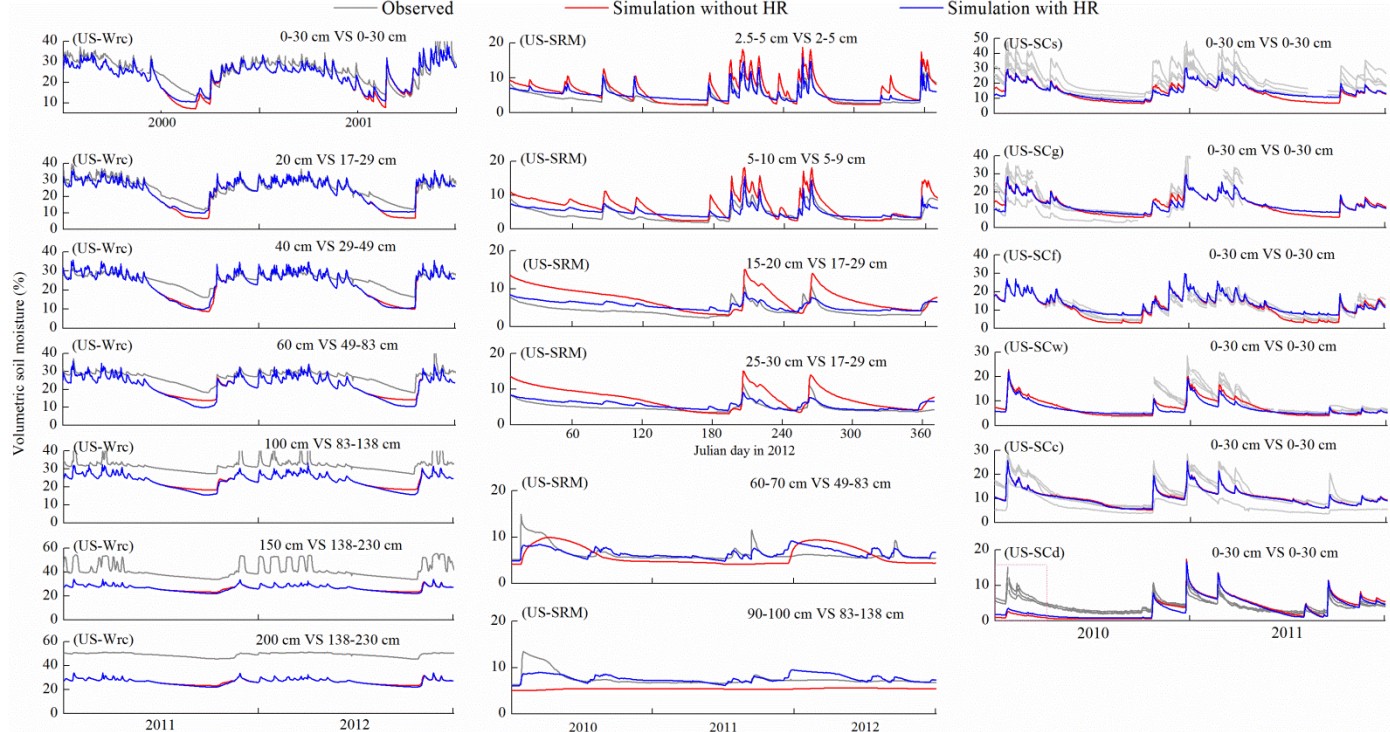

**Fig. 1**. Observed and simulated soil moisture over selected years. Labels at the upper right corner of each soil moisture panel show the depths of observed and simulated soil moisture. For example, "20 cm VS 17-29 cm" means the observation depth of soil moisture is 20 cm and the simulated results at depths of 17-29 cm were compared with this observation. Within panels for southern California sites (US-SCs, US-SCg, US-SCf, US-SCw, US-SCc, and US-SCd), the four grey lines are data from the four CS-616 soil moisture sensors at 0-30 cm depth. Results were shown for the last two years only for the above 30 cm depth at US-SRM site for clarity. The rectangular box indicated period with incomplete precipitation record.




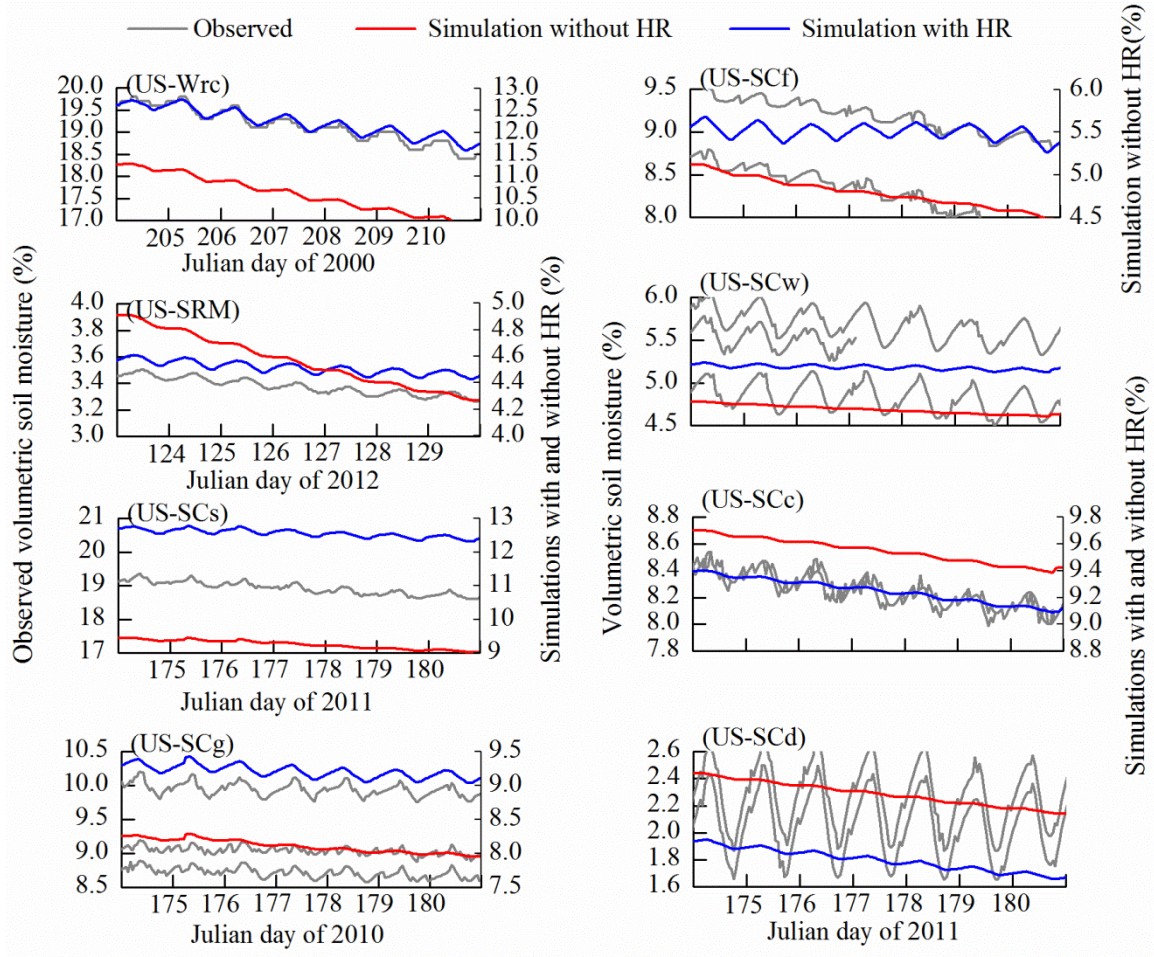

**Fig. 2**. Observed and simulated soil moisture for depth of 0-30 cm during dry periods.




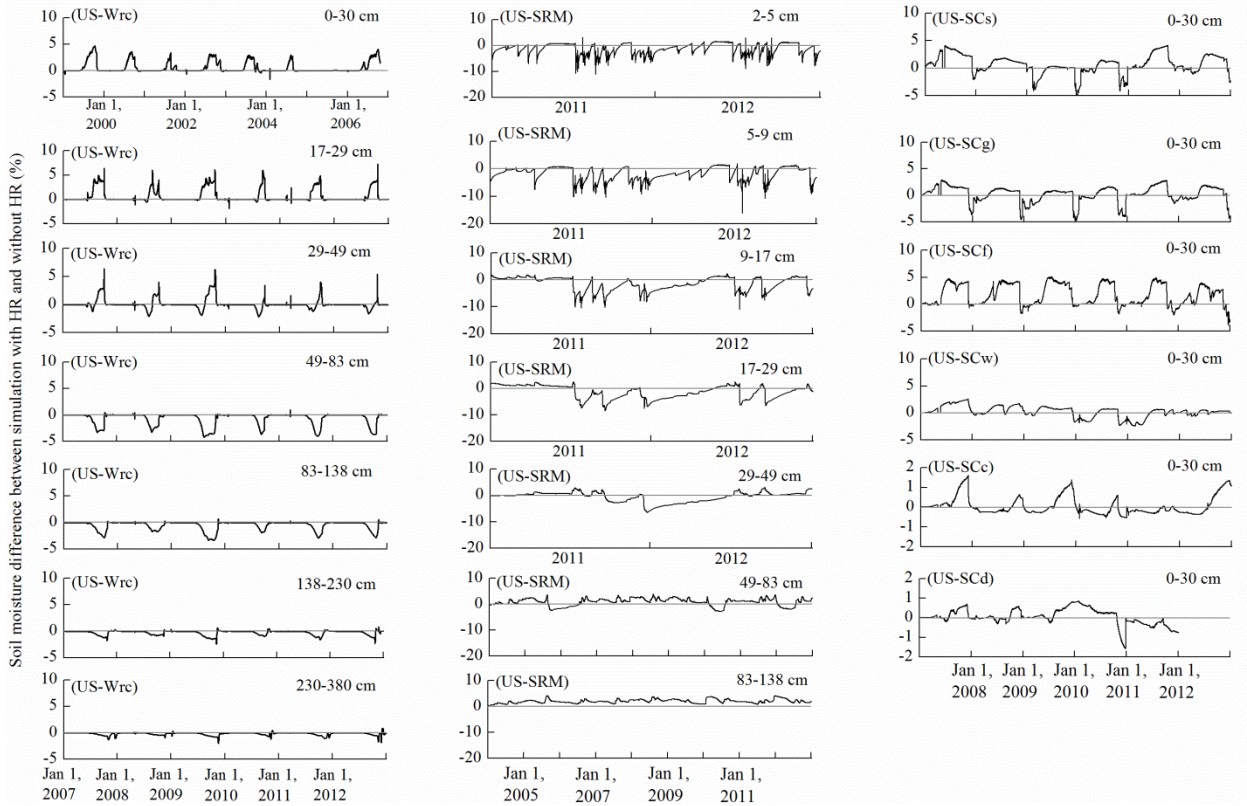

**Fig. 3**. Soil moisture difference between simulations with HR and without HR at each site. Labels at the upper right corner of each soil moisture panel show the depths of simulated soil moisture.







**Fig. 4.** Modeled hydraulic redistribution (HR)-induced change in volumetric soil moisture at the eight study sites. Results shown here are the averaged values for Julian days over the entire simulation period.



**Fig. 5**. Observed and simulated daily evapotranspiration at the eight study sites.




**Fig. 6**. Observed and simulated hourly evapotranspiration (ET) and its components at the eight study sites. "QSOIL" is ground evaporation, "QVEGE" is canopy evaporation, and "QVEGT" is transpiration. "Observation" represents observed evapotranspiration. "Dry period" and "wet period" represent average values over dry season and wet season (defined in Table 5), respectively.





Fig. 7. Observed and simulated weekly Bowen ratio during dry periods.




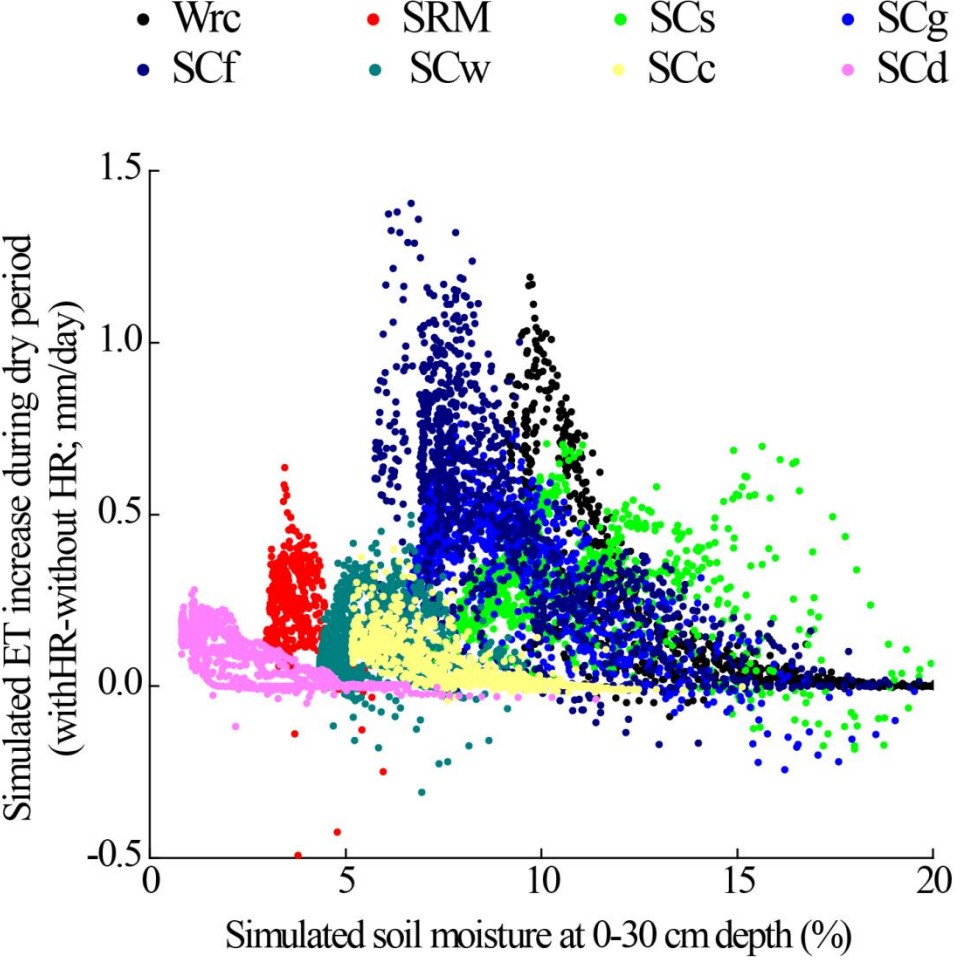

**Fig. 8.** Simulated changes in evapotranspiration (ET) caused by hydraulic redistribution (HR) as a function of soil moisture at all eight sites.







**Fig. 9** Sensitivity of simulated contribution of HR to ET (defined in Table 5) to selected parameters in Ryel et al.'s 2002 equation describing HR. Circled parameter set was used in Table 5.