# Peer review of "Combined measurement and modeling of the hydrological impact of hydraulic redistribution using CLM4.5 at eight AmeriFlux sites"

_Hydrology and Earth System Sciences, 2016_

## Referee Comment (RC1) · Anonymous Referee #1 · 23 Jan 2016

This paper by Fu et al investigated an interesting question about the important effects of hydraulic redistribution on soil moisture, evapotranspiration, Bowen ratio across eight Ameriflux sites characterized by contrasting climate regimes and multiple vegetation types. To address this question, the authors incorporated Ryel et al.'s (2002) empirical equation describing HR into the NCAR Community Land Model Version 4.5 (CLM4.5). They mainly found that HR is a significant hydraulic flow in wet sites with seasonal dry climate and inclusion of HR into CLM4.5 improved the model-measurement match in soil moisture, evapotranspiration, Bowen ratio particularly during dry seasons. The merit of this study is the integration of empirical data into CLM 4.5 across eight Ameriflux sites characterized by contrasting climate regimes and multiple vegetation types,

although there are still uncertainty in some parameters. Overall, this paper is a pleasure to read and the results and conclusion are convincing.

Here I have some suggestions to the authors that may help improve the paper: 1) NCAR Community Land Model Version 4.5 (CLM4.5) is a big component in this study. The authors may need to give more information about CLM4.5 by summarizing the main characteristic of CLM4.5. 2) HR could be expressed in different spatial scale (i.e., patch scale and/or landscape scale). The authors may need to clarify this point. Also recent studies show that HR in cases with groundwater are significantly in contrast to the cases without ground water. However, at this time the information about the root depth/distribution, the potential access to ground water is lacked. 3) Indeed, there are some studies which have simulated the magnitude of HR flux itself and also the effects of HR on ET and vegetation. Thus, the authors may change the focus of novelty in this study. Would the novelty of this study focus on the comparison in the eight Ameriflux sites characterized by contrasting climate regimes and multiple vegetation types? Especially the six new sites along the Southern California Climate Gradient where HR has been less investigated? The integration of empirical data into CLM 4.5? 4) HR could also affect vegetation photosynthesis, growth, and dynamics. The results of vegetation dynamics as affected by HR may need to be included. There are also potentially two way interactions between vegetation and HR. 5) The modeling simulations tend to overestimate HR as compared to field studies. There are some studies which argue the dynamics root uptake (compensation) and plant water storage by stems could undermine the magnitude of HR. These points may need to be recognized in some details in the discussion.

Other minors points: Abstract (page 1): there are some sentences which are very similar with those in the main text. The authors may want to rephrase these sentences. line 25: it may be better to specify what the model-measurement match are. Introduction line 1 (page 2): delete soil moisture content since you have already talked about soil water potential gradient. May add isotope since isotope is another major method.

[Figure]

line 4: may delete "following a precipitation event". Hydraulic descent generally occur after rainfall events, but not necessarily since in theory hydraulic descent occur as long as soil water potential in the shallow soil is higher than that in deep soil. Line 12: delete "found" and also two "," Line 13: may need to give more information about dynamics root uptake since some readers (especially the beginners studying HR) may not know this well. Line 15: again, the authors may need to change the novelty of this study. Line 17: the authors may need to add more information about "compensating for other hydrological deficiencies in the default model". I guess that not too many readers as beginners studying HR know about this. Line 25: add "that" between "show" and "trees and shrubs..." Page 3 Line 1: change "use modeling approach, the Ryel et al. (2002) approach," to "use modeling approach by Ryel et al. (2002)" Line 2: delete "from which HR can be directly inferred"? since it is obvious that the magnitude of HR flux is determined by soil water potential gradient (soil moisture) based on Ryel et al. (2002) approach Page 4 Line 9: what is the time scale of vegetation dynamics in CLM4.5. Or do you keep vegetation dynamics static in the model? Line 27: change "for" to "because of" Page 6 The results about figure 1: some points may need to be clarified. First: is the soil moisture reported as the daily or 30 min value? Second: when I analyzed the soil moisture value in the shallow and deep soil with and without HR for US-SRM, there is clearly hydraulic descent. But when looking at the soil moisture value in the shallow and deep soil with and without HR for US-Wrc, the red line (without HR) is consistently lower than the blue line (with HR) for some period (may be dry seasons when HR is significant), which means that hydraulic lift (HL) occurs across all the soil profile (0-230 cm). This is in contrast to what is shown in figure 3. Page 7 Line 4: change Table 6 to Table 5? Since the authors showed Table 6 before Table 5 in the result section Line 18: sap flow measurements are usually done for individual trees (patch scale). The authors may need to check whether Scott et al (2008) report the values of HR for individual trees (patch scale) or landscape scale. Line 25: the authors may want to present the results to appendix. Page 10 Line 16: while doing sensitivity analysis, the range of CRT (0.1-1.5) seems to be large with an

order of magnitude. Did the author try other values with narrower range (say 0.4-0.8)? The study "Modeled hydraulic redistribution in tree-grass, CAM-grass, and tree-CAM associations: the implications of Crassulacean Acid Metabolism (CAM)." shows that HL is not sensitive to CRT with narrower range. Line 23: change "Measurement and modeling both" to "Modeling simulations"? since HR is not directly measured in this study. Page 11 Line 8-10: change "The US-Wrc panel in Fig. 4 also shows…..... (It is worth noting that the CLM4.5+HR model does not include the temperature fluctuation-driven vapor transport within soil shown by Warren et al. (2011)" to "The US-Wrc panel in Fig. 4 also shows….....,  although the CLM4.5+HR model does not include the temperature fluctuation-driven vapor transport within soil shown by Warren et al. (2011)"? Line 12: change 2 m to 2000 mm Line 13-15: change the sentence "Hydraulic descent is limited, averaging 5.0 mm H2O yr-1 during 1999-2012, perhaps because soil moisture is higher with depth, limiting the driving gradient for hydraulic descent. " to "Hydraulic descent is limited with the average values being 5.0 mm H2O yr-1 during 1999-2012"? If soil moisture is higher with depth, HL would occur. Line 25: may need add reference about bedrock if it is available. Page 13 Line 2-4: the authors may need to give more information about "deep water uptake" (Markewitz et al., 2010), "HR representation models" (Amenu and Kumar, 2008; Quijano and Kumar, 2015) or rephrase "deep water uptake" and "HR representation models". At this time, there are not clear.

---

## Referee Comment (RC2) · Anonymous Referee #2 · 24 Feb 2016

**Referee comments on "Combined measurement and modeling of the hydrological impact of hydraulic redistribution using CLM4.5 at eight AmeriFlux sites" by C. Fu et al.**

**General Comments**

This study investigated the HR process and its effects on land surface water and energy cycles in a comprehensive manner by using the modeling method. The Ryel HR scheme (Ryel et al., 2002) was included in the CLM4.5 land surface model, and the new model (CLM4.5+HR) was applied at eight AmeriFlux sites with various climates, vegetation types and soil types. Quantitative analyses showed that including HR could improve the modeling of soil moisture, ET and Bowen ratio, which suggested that HR could be an important process, in many circumstances, particularly in environments where the annual precipitation was not very low and the temporal distribution of precipitation was quite uneven. Based on the numerical simulations using CLM4.5 and CLM4.5+HR at the eight sites, HR flux and the effects of HR on land surface water and energy budgets in diverse ecosystems were quantified and analyzed at different aspects. The sensitivities of model results to the important parameters of the Ryel HR scheme were also examined for the sites with varied climates.

This manuscript addresses relevant scientific questions within the scope of HESS. The methods are clearly described. The conclusions are based on well-designed numerical simulations and detailed analyses. The tables and figures are well made.

Some minor revisions are needed to make the presentation more precise or concise.

**Specific Comments**

1. In this manuscript, it is stated that the Ryel HR scheme (Ryel et al., 2012) is used. However, there are some differences between the Equation 1, which representing the HR process, of this manuscript and the original equation (i.e., Equation 6 in Ryel et al., 2012): (1) This manuscript: $c_j$, the factor reducing soil-root conductance in the giving soil layer $j$; Ryel et al.: $\max(c_i, c_j)$, the larger one of the two factors in the receiving soil layer $i$ and the giving soil layer $j$; (2) In the denominator: This manuscript: $F_{root}(j)$, the root fraction in the giving soil layer $j$; Ryel et al.: This parameter can be $F_{root}(i)$ or $F_{root}(j)$, which depends on the comparison between soil moisture of the layer $i$ and that of the layer $j$; (3) This manuscript: It is not explicitly stated whether HR occurs during daytime in the modeling; Ryel et al.: It is stated that HR is "turned off" during daytime in the modeling of their study. These aspects can be explained in this manuscript.

In addition, if the definition of $C_{RT}$ here is same as that of Ryel et al., "of the entire active root system for water" needs to be added after "soil-root conductance" in Page 4 Line 19.

2. Root distribution along the vertical direction in the soil column is an important input in the HR modeling. In Page 4 Line 20, it is mentioned that the root distribution is based on Zeng (2001). More information about root distribution can be provided (e.g., how is the root distribution represented; Root-distribution differences between the eight AmeriFlux sites; etc.). This information can be helpful for the readers to understand the simulation results.

**Technical Corrections**

Page 1 Line 18: Add "land" between "on" and "surface water".

Page 1 Line 19: Replace "it" with "the impact".

Page 2 Lines 8-9: Remove "do so by".

Page 2 Line 9: Replace "incorporating" with "incorporate".

Page 2 Line 9: Add "e.g.," before "Ryel et al., 2002" (because some studies listed did not use the Ryel HR scheme, for example, Lee et al., 2005).

Page 2 Line 24: Replace "CA" with "California".

Page 2 Line 30: Replace "2m of soil" with "2-m soil layer".

Page 3 Line 1: Replace "objective" with "objectives"; Replace "is" with "are".

Page 3 Lines 1-2: Replace "modeling approach, the Ryel et al. (2002) approach," with "HR scheme (*or model ?*) (Ryel et al., 2002)".

Page 3 Line 3: Add "land" between "on" and "surface water"; Replace "it" with "the impact".

Page 3 Line 4: Rephrase "This is done through incorporating Ryel et al.'s (2002) simple empirical equation for HR flux" to be "For these objectives, we incorporated the Ryel HR scheme".

Page 3 Line 5: Add "(CLM4.5)" after "4.5"; Replace "applying" with "applied"; Remove "the" before "eight AmeriFlux sites".

Page 3 Line 20: Replace "degrees C" with "$°C$".

Page 4 Line 12: Replace "quantify" with "represent"; Replace "Ryel et al.'s (2002) equation" with "the Ryel HR scheme (Ryel et al., 2002)".

Page 4 Lines 12-14: Rephrase "This equation has been widely used in HR modeling studies (Lee et al., 2005; Zheng and Wang, 2007; Wang, 2011; Li et al., 2012) and its variations (e.g. Yu and D'Odorico, 2015)." to be "Many HR modeling studies used this HR scheme (e.g., Zheng and Wang, 2007; Wang, 2011; Li et al., 2012) or its variations (e.g., Lee et al., 2005; Yu and D'Odorico, 2015).".

Page 4 Line 25: Rephrase "The relationship between root hydraulic conductivities and soil moisture in equation (2) is similar to ..." to be "The relationship of Equation 2 is similar to ...". (because the description is not accurate)

Page 4 Line 28: Replace "equation 1" with "Equation 1".

Page 5 Line 1: Replace "MPb" with "MPa".

Page 5 Line 2: Replace " "Ryel et al. 2002" equation" with "Ryel HR scheme".

Page 5 Line 6: Add "Clapp and Hornberger" between "influences of" and ' "B" '.

Page 5 Line 6: Replace "based on" with "in the".

Page 5 Line 11: Replace ' "B"s ' with ' "B" values '; Replace "lower" with "deeper".

Page 5 Lines 12-13: Rephrase "... identical parameter "B" tuned ..." to be "... the identical "B" value tuned ...".

Page 5 Lines 16-17: Remove "for the sites".

Page 5 Line 23: The meaning of "conservatively" is not clear.

Page 5 Lines 25-26: "... in surrounding portions of the signal trace ..." is not clear.

Page 5 Line 26: Add "existence of" before "HR".

Page 6 Line 4: Replace "increment" with "soil layer".

Page 6 Line 12: Add " – " after "the models".

Page 6 Lines 20-21: Remove the comma " , "; Remove "US-SRM"; Add "at this site" after "... 90-100 cm depths".

Page 6 Line 22: Replace "rain" with "soil water"; Replace "surface" with "shallow"; Rephrase "it could be delivered by percolation alone" to be "the percolation process".

Page 6 Line 31: The "hydrological effect" was "far lower" than what?

Page 7 Line 14: Add "flux" after "HR".

Page 7 Line 16: Replace "... the magnitude of ..." with "... the magnitude in the ...". (to make this sentence more readable)

Page 7 Line 19: The hydraulic descent magnitude (12 – 38 mm/day and 35 mm/day) seems to be very large. Please verify these values.

Page 7 Line 25: "..., the maximum depth of HR-induced soil moisture increases ..." is not clear.

Page 7 Lines 28-29: "... used soil suction to roughly control the magnitude of HR ..." is not clear.

Page 8 Line 3: Remove "+HR" after "CLM4.5"; Replace "percent" with "percentage points".

Page 8 Line 5: Add "those of" before "decreased soil moisture".

Page 8 Line 8: Remove "at the six Southern California US-SC sites".

Page 8 Line 11: Add "in" before "2011".

Page 8 Line 13: Replace "Ryel et al.'s (2002) HR model" with "the Ryel HR scheme".

Page 8 Line 14: Remove "up to 1%"; Add " (up to 1%) " after "soil moisture".

Page 8 Line 15: Remove "up to 4%"; Add " (up to 4%) " after "soil moisture".

Page 8 Line 16: Remove "%"; Add " (in the unit of %) " after "soil moisture change".

Page 8 Line 18: Switch "site" and "US-Wrc".

Page 8 Line 19: Switch "HR" and "downward".

Page 8 Lines 23-24: "... , a gradient in the depth and temporal extent of HR on modeled soil moisture was clear." needs to be rephrased.

Page 8 Line 27: Add "range" after "depth"; Replace "influence of HR" with "HR influence".

Page 8 Line 29: Add "$H_2O$" between "mm" and "$d^{-1}$". (to be consistent with the same unit in this paragraph)

Page 8 Lines 30-32: It is not clear these results are from a specific year, or average values of multiple years?

Page 9 Line 4: Replace "varying" with "various".

Page 9 Line 7: Remove "also improvement".

Page 9 Line 9: Remove " (hourly) ".

Page 9 Line 9: These results are average values of one year, or multiple years?

Page 9 Line 10: Replace "tends" with "tended".

Page 9 Line 10: Remove "observed mid-day". (These words are redundant.)

Page 9 Lines 10-11: Fig. 6 shows that, for most circumstances, the modeled high ET values from CLM4.5+HR are closer to the observations, as compared to those from CLM4.5. This point can be put forward around here.

Page 9 Lines 17-18: "... was subsequently taken up from deeper layers by plants during transpiration." – But soil water in deep layers could also be transferred to the shallow soil via the upward HR process when plant transpiration rate is low.

Page 9 Line 21: Move "(0.47 mm $H_2O$ $d^{-1}$)" to after "US-SCf site"; Remove "corresponding ET increase is".

Page 9 Line 22: Add "site" after "US-SRM".

Page 9 Line 25: Replace "Ratio" with "ratio".

Page 9 Line 32: Rephrase "synthesizes" to be "reflects the integration of".

Page 10 Line 5: Rectify "hear" to be "heat".

Page 10 Line 8: "... beyond those studied here" is not clear.

Page 10 Line 10-11: Rephrase "... HR-induced increases in ET are maximal at sites with mid-range soil moistures" to be " ... HR-induced large increases in ET primarily occur at sites with mid-range soil moistures". (because in Fig. 8, for sites of moderate soil moisture, some HR-induced increases of ET are small)

Page 10 Line 14: Replace ' "Ryel et al. 2002" equation ' with 'Ryel HR scheme'.

Page 10 Line 17: "at the height of HR" is not clear.

Page 10 Line 17: Add "during the periods with high HR flux". (because in Fig. S10, hydraulic lift magnitude could double when HR flux is moderate or low)

Page 10 Line 27: Add "as" after "considered".

Page 11 Line 7: Replace "yr" with "year".

Page 11 Line 13: Replace "dominated" with "primarily caused".

Page 11 Line 14: Replace "H2O" with "$H_2O$".

Page 11 Line 17: Add "soil" before "water content".

Page 11 Line 20: Replace "is" with "was".

Page 11 Line 25: Add "missing representation of" before "hydraulic lift".

Page 11 Line 31: Remove "the monitored".

Page 12 Line 2: Rephrase ", including at sites modeled here" to be "including those conducted at sites of this study".

Page 12 Line 3: Replace "contributed" with "supplied"; Replace "to" with "of"; Rephrase "upper 2 m of the soil" to be "top 2-m soil layer".

Page 12 Line 5: Replace "similar" with "comparable". ( The meaning of 28% is different from that of 32%: 28% = HL / ET from the top 2-m layer; 32% = HL / ET from the whole soil column. )

Page 12 Lines 6-7: Add "transpiration of the" after "estimated"; Remove "transpiration" before "in 2004".

Page 12 Line 21: Replace "are" with "were".

Page 12 Line 28: Replace "idea" with "hypothesis" or "assumption".

Page 13 Line 3: The reference "Quijano and Kumar, 2015" is missing in the reference list.

Page 13 Lines 8-10: These hydraulic lift results are maximum values in a time period? Or mean values of some periods?

Page 13 Line 14-16: These ET increases are maximum values in a time period? Or mean values of some periods?

Page 13 Line 23: Replace "Ryel et al. 2002 equation" with "Ryel HR scheme".

Page 13 Line 25: Add "during the periods with high HR flux" after "hydraulic lift".

Page 13 Lines 25-26: Add "larger change of" before "hydraulic descent"; Remove "was more sensitive".

Page 13 Lines 29-31: "HR has been confirmed in many ecosystems" does not convincingly support the argument that "HR should be included for all ecosystems". Need some revising here.

Page 13 Line 32: Replace "a large number of" with "eight". (Eight is not a large number.)

Page 18 Lines 3, 6: Replace "Ryel et al.'s 2002 equation describing HR" with "the Ryel HR scheme".

Page 19: Columns 1 and 2: it is better to align left.

Page 20: Column 5 Row 3: Add "m" after "0.25-5".

Page 22 Line 1: Replace ' "Ryel et al. 2002" HR model ' with "Ryel HR scheme".

Page 22 Line 2: Add "of the entire active root system for water" after the first "conductance", if the definition of $C_{RT}$ is same as that in Ryel et al. (2002).

Page 22 Line 3: Remove the quotation marks " " around $b$.

Page 23: Need to specify the results of Table 5 are mean values of one year or mean values of multiple years.

Page 23: The last column: these values need to be multiplied by 100.

Page 24: The 4[th] line from the bottom: Replace "soil moistures" with "soil moisture data".

Page 24: The 2[nd] line from the bottom: Remove "RMSE for". (They are redundant.)

Page 24: The last line: Rephrase "improved model fit to data" to be "a better fit between model results and observed data".

Page 25: The 2[nd] Line from the bottom: Replace "at 0-30 cm depth" with "at different depths in the top 30-cm soil layer".

Page 25: The last two lines: "Results were shown for the last two years only for the above 30 cm depth at US-SRM site for clarity." is not clear. This sentence seems to mean that: "For the top 30-cm soil layer at the US-SRM site, only the results of the last year (Year 2012) were shown for clarity."

Page 25: The last line: "The rectangular box" is missing in the figure.

Page 27: The last line:  Remove the first "soil moisture".

Page 28: The units are missing in Fig. 4.

Page 29 Lines 1 and 2: Remove "(mm)" in the legend.

Page 30: In the ET plots, the meaning of the thin vertical lines is not described.

Page 30: The 2nd line from the bottom: "dry  season  and  wet  season" are from one year or multiple years.

Page 31: The results of Year 2004 at the US-SRM site are contrary to the other results, which can be discussed in the text.

Page 33: Replace " $\phi_{50}$ " with " $|\varphi_{50}|$ ".

Page 33: Units of $C_{RT}$ and $\varphi_{50}$ are missing.

Page 33: The last two lines: Replace "Ryel et al.'s 2002 equation describing HR" with "the Ryel HR scheme".

---

## Author Comment (AC1) · 1 Mar 2016

Please check the attachments for the revised manuscript and our responses to comments, thanks much for your comments!

Please also note the supplement to this comment:
http://www.hydrol-earth-syst-sci-discuss.net/hess-2016-24/hess-2016-24-AC1-supplement.zip

---

## Author Response (AR1)

We thank the reviewer for his/her diligent effort to help improve the manuscript. In the following please find our point-by-point response to these comments, marked with a "R:"

5    1) NCAR Community Land Model Version 4.5 (CLM4.5) is a big component in this study. The authors may need to give more information about CLM4.5 by summarizing the main characteristic of CLM4.5.

R: The following texts have been added in the revised manuscript to describe the characteristic of CLM:

"Surface heterogeneity in CLM is represented using a nested hierarchy of grid cells, land units, snow/soil columns, and plant functional types (PFTs). Different PFTs differ in physiological, structural and biogeochemical parameters. Within each
10   vegetated land unit, multiple columns can exist, and multiple PFTs can share a column; vegetation state variables, surface mass and energy fluxes are solved at the PFT level, and soil parameters and processes are solved at the column level. Surface fluxes at the grid cell level (e.g., ET) are the area-weighted average across different components (PFTs, columns, and land units)."

2) HR could be expressed in different spatial scale (i.e., patch scale and/or landscape scale). The authors may need to clarify
15   this point. Also recent studies show that HR in cases with groundwater are significantly in contrast to the cases without ground water. However, at this time the information about the root depth/distribution, the potential access to ground water is lacked.

R: For the scale, we have explained the routine practice in applying land surface models to a site and try to avoid calling it a specific scale, namely, "At each study site, the simulations were implemented for the footprint of eddy flux tower" (lines 5-
20   6, page 4).

The following sentences have been added to illustrate the root depth/distribution information and the potential access to groundwater:

"There are ten active soil layers in CLM, and a maximum depth of 3.8 m is used in this study (Table 3). The PFT-level root fraction $r_i$ in each soil layer is,

$$r_i = \begin{cases} 0.5 \cdot [exp(-r_a z_{i-1}) + exp(-r_b z_{i-1}) - exp(-r_a z_i) + exp(-r_b z_i)] & for \ \ 1 \le i < 10 \\ 0.5 \cdot [exp(-r_a z_{i-1}) + exp(-r_b z_{i-1})] & for \ \ \ \ \ i = 10 \end{cases} \tag{1}$$

where $z_i$ is the depth at the bottom of soil layer $i$, and $z_0$ is zero. The PFT-dependent root distribution parameters $r_a$ and $r_b$ are adopted from Zeng (2001). From Equation (1), $r_i$ decreases exponentially with depth. In the present study, roots did not have access to groundwater through the simulation periods at all sites except US-Wrc where groundwater could rise into the tenth soil layer during the wet season. However, the groundwater level was below the tenth soil layers during dry season when HR
30   occurred at the US-Wrc site as shown in Results section 3.1.2" (lines 8-16, page 4).

3) Indeed, there are some studies which have simulated the magnitude of HR flux itself and also the effects of HR on ET and vegetation. Thus, the authors may change the focus of novelty in this study. Would the novelty of this study focus on the comparison in the eight Ameriflux sites characterized by contrasting climate regimes and multiple vegetation types? Especially the six new sites along the Southern California Climate Gradient where HR has been less investigated? The
35   integration of empirical data into CLM 4.5?

R: To clarify this, we revised the corresponding sentences in abstract and introduction sections:

Changed from: 'few (if any) has tackle the magnitude of the HR flux itself or the soil moisture dynamics from which HR magnitude can be directly inferred', to: 'few (if any) has done cross-site comparisons for contrasting climate regimes and multiple vegetation types via the integration of measurement and modeling'.

5  In introduction section:

Changed from: "However, most of these studies focused on how including HR might improve the model performance in simulating ET and in some cases soil moisture, and few (if any) has tackled the magnitude of the HR flux itself or the soil moisture dynamics from which HR magnitude can be directly inferred. It is not clear from these previous studies whether the HR-derived model performance might be caused by HR compensating for other hydrological deficiencies in the default

10  model" to "Currently, few (if any) has investigated the effects of HR on land surface water and energy cycles in a comprehensive manner by using both the monitoring and modeling methods for contrasting climate regimes and multiple vegetation types";

Changed from: 'The objective of this study is to examine the performance of a commonly used modeling approach, the Ryel et al. (2002) approach, in capturing the magnitude of HR flux and/or soil moisture dynamics from which HR can be directly

15  inferred', to: 'The objectives of this study are to investigate the impact of HR on land surface water and energy budgets based on both observational data and numerical modeling, and to explore how the impact may depend on climate regimes and vegetation conditions. Observed soil moisture at the six Southern California Climate Gradient sites was corrected for temperature first, and then HR signal was checked using the wavelet method. The modeling investigation is done through incorporating the HR scheme of Ryel et al. (2002) into the NCAR Community Land Model Version 4.5 (CLM4.5). To apply

20  the hybrid model to the eight AmeriFlux sites, we first examined the performance of the hybrid model in capturing the magnitude of HR flux and/or soil moisture diel fluctuation from which a reasonable HR flux magnitude can be directly inferred; we then analyzed the role of HR in the water and energy cycles. The sensitivity of the modelled HR to parameters and the uncertainty in the modeling were also investigated in the present study. (lines 1-9, page 3)

4) HR could also affect vegetation photosynthesis, growth, and dynamics. The results of vegetation dynamics as affected by

25  HR may need to be included. There are also potentially two way interactions between vegetation and HR.

R: Vegetation dynamics is simulated in our following-up study that investigates the influences of HR on plant growth and more importantly on carbon/nitrogen cycles. Interestingly, for the HR impact on surface water and energy budgets, the results based on dynamic vegetation (which are being included as supplemental materials for our follow-up paper on carbon and nitrogen dynamics) are qualitatively similar. In the present manuscript however, we stick to the prescribed LAI so that

30  the model and observational data pertain to the same vegetation conditions. However, in response to this comment, we have added the rationale of using prescribed LAI:

'More subtle and interesting sources of uncertainty also likely influenced the model-measurement match. For example, strong inter-annual variation of precipitation, fire, and recovery from fire caused rather abrupt changes of PFT coverage and LAI at the US-SCs site. The US-SCg site is undergoing restoration to a native grassland community, and a large community

of ephemeral annuals comes up following winter or summer rains at the US-SCc site. These variations were difficult to capture by satellite remote sensing data but undoubtedly affected soil moisture and ET in interesting ways. Without detailed ground-observational data to quantify them, simulations in this study used a climatological LAI seasonal cycle'. (lines 7-13, page 13)

5) The modeling simulations tend to overestimate HR as compared to field studies. There are some studies which argue the dynamics root uptake (compensation) and plant water storage by stems could undermine the magnitude of HR. These points may need to be recognized in some details in the discussion.

R: we had calibrated and analyzed the magnitude of the HR flux in the present study, as shown in the method (lines 14-20, page 5) and results section (section '3.1.1 HR flux simulations').

The influences of the dynamic root uptake and plant water storage by stems on HR simulation have been mentioned, and two more references have been added in the discussion section in the revised manuscript, as

"In addition, the effects of several important factors warrant further investigation, including for example the root architecture (Yu and D'Odorico, 2014), dynamic root water uptake (Zheng and Wang, 2007), deep tap roots (Markewitz et al., 2010), above ground storage capacity (Hultine et al., 2003), temperature fluctuation-driven vapor transport within soil (Warren et al., 2015), and macro-pore flow (Fu et al., 2012,14). It is also important to compare different representations of HR models (Amenu and Kumar, 2008; Quijano and Kumar, 2015) to examine uncertainties related to model structure.  (lines 23-28, page 13)

Other minors points: Abstract (page 1): there are some sentences which are very similar with those in the main text. The authors may want to rephrase these sentences.

R: we have done some slight revisions to the abstract.

line 25: it may be better to specify what the model-measurement match are.

R:  It is now specified that the match is for "evapotranspiration, Bowen ratio, and soil moisture dynamics".

Introduction line 1 (page 2): delete soil moisture content since you have already talked about soil water potential gradient. May add isotope since isotope is another major method.

R: Some former studies did monitor soil water potential or soil moisture only, so we decided to keep both variables.  Isotope method and corresponding reference have been added.

line 4: may delete "following a precipitation event". Hydraulic descent generally occur after rainfall events, but not necessarily since in theory hydraulic descent occur as long as soil water potential in the shallow soil is higher than that in deep soil.

R: We have specified it to be "usually following a precipitation event" (line 5, page2).

Line 12: delete "found" and also two ","

R: "found" and corresponding colon have been deleted in the revised manuscript.

Line 13: may need to give more information about dynamics root uptake since some readers (especially the beginners studying HR) may not know this well. Line 15: again, the authors may need to change the novelty of this study.

R: The following annotations have been added to illustrate the dynamic root uptake, "(preferential uptake of moisture from areas of the root zone where moisture is more available, Lai and Katul, 2000)". (lines 15-16, page2)

Line 17: the authors may need to add more information about "compensating for other hydrological deficiencies in the default model". I guess that not too many readers as beginners studying HR know about this.

R: we have deleted this sentence during rephrasing the novelty of the study.

Line 25: add "that" between "show" and "trees and shrubs: : :"

R: Added.

Page 3 Line 1: change "use modeling approach, the Ryel et al. (2002) approach," to "use modeling approach by Ryel et al. (2002)"

R: we have deleted this sentence during rephrasing the novelty of the study.

Line 2: delete "from which HR can be directly inferred"? since it is obvious that the magnitude of HR flux is determined by soil water potential gradient (soil moisture) based on Ryel et al. (2002) approach

R: What was 'inferred" here is the validity of the magnitude of HR, so we are apt to keep it unchanged.

Page 4 Line 9: what is the time scale of vegetation dynamics in CLM4.5. Or do you keep vegetation dynamics static in the model?

R: In the revised manuscript, we have added: 'The plant growth and carbon / nitrogen cycles were not simulated in this study. Instead, LAIs for each PFT were prescribed based on observational data'. (lines 1-2, page4)

Line 27: change "for" to "because of"

R: Changed to "due to".

Page 6 The results about figure 1: some points may need to be clarified. First: is the soil moisture reported as the daily or 30 min value? Second: when I analyzed the soil moisture value in the shallow and deep soil with and without HR for US-SRM, there is clearly hydraulic descent. But when looking at the soil moisture value in the shallow and deep soil with and without HR for US-Wrc, the red line (without HR) is consistently lower than the blue line (with HR) for some period (may be dry seasons when HR is significant), which means that hydraulic lift (HL) occurs across all the soil profile (0-230 cm). This is in contrast to what is shown in figure 3.

R: 'at daily time scale' has been added to illustrate the time scale in Fig. 1. (line 22, page 6)

Actually, the red line (without HR) is NOT consistently lower than the blue line (with HR). The red line (without HR) was lower than the blue line (with HR) above 29 cm and was higher than the blue line below "49 cm" during dry periods at US-Wrc site, illustrating hydraulic lift from below 49 cm to above 29 cm, and it was consistent with what was shown in Fig. 3.

Page 7 Line 4: change Table 6 to Table 5? Since the authors showed Table 6 before Table 5 in the result section

R: The sequence of the Tables has been corrected.

Line 18: sap flow measurements are usually done for individual trees (patch scale). The authors may need to check whether Scott et al (2008) report the values of HR for individual trees (patch scale) or landscape scale.

R: Scott et al (2008) reported the values of HR for landscape scale/ecosystem scale, and we have added "at ecosystem scale" in the revised manuscript. (line 8, page 8)

Line 25: the authors may want to present the results to appendix.

R: we have presented the results as Fig. S9 in appendix.

Page 10 Line 16: while doing sensitivity analysis, the range of CRT (0.1-1.5) seems to be large with an order of magnitude. Did the author try other values with narrower range (say 0.4-0.8)? The study "Modeled hydraulic redistribution in tree-grass, CAM-grass, and tree-CAM associations: the implications of Crassulacean Acid Metabolism (CAM)." shows that HL is not sensitive to CRT with narrower range.

R: As shown in Fig. S10, the modeled HL was sensitive to CRT even with a narrow range, such as 0.5-0.75.

Line 23: change "Measurement and modeling both" to "Modeling simulations"? since HR is not directly measured in this study.

R: we have changed "Measurement and modeling both" to "The cross-ecosystem comparisons" in the revised manuscript.

Page 11 Line 8-10: change "The US-Wrc panel in Fig. 4 also shows: : :... (It is worth noting that the CLM4.5+HR model does not include the temperature fluctuation-driven vapor transport within soil shown by Warren et al. (2011)" to "The US-Wrc panel in Fig. 4 also shows: : :..., although the CLM4.5+HR model does not include the temperature fluctuation-driven vapor transport within soil shown by Warren et al. (2011)"?

R: The contents in parenthesis are for information only, and are not closely related to the preceding sentence. So no change has been made.

Line 12: change 2 m to 2000 mm

R: Changed as suggested.

Line 13-15: change the sentence "Hydraulic descent is limited, averaging 5.0 mm H2O yr-1 during 1999-2012, perhaps because soil moisture is higher with depth, limiting the driving gradient for hydraulic descent. " to "Hydraulic descent is limited with the average values being 5.0 mm H2O yr-1 during 1999-2012"? If soil moisture is higher with depth, HL would occur.

R: Changed as suggested.

Line 25: may need add reference about bedrock if it is available.

R: 'as shown in Kitajima et al. (2013)' has been added in the revised manuscript.

Page 13 Line 2-4: the authors may need to give more information about "deep water uptake" (Markewitz et al., 2010), "HR representation models" (Amenu and Kumar, 2008; Quijano and Kumar, 2015) or rephrase "deep water uptake" and "HR representation models". At this time, there are not clear.

R: About deep water uptake, we have changed it to 'deep tap roots (Markewitz et al., 2010); we have also changed "HR representation models" to "It is also important to compare different representations of HR models …" in the revised manuscript. (lines 25-27, page 13)

**Responses to reviewer#2**

We thank the reviewer for his/her diligent effort to help improve the manuscript. In the following please find our point-by-

10  point response to these comments, marked with a "R:"

Specific Comments

1. In this manuscript, it is stated that the Ryel HR scheme (Ryel et al., 2012) is used. However, there are some differences between the Equation 1, which representing the HR process, of this manuscript and the original equation (i.e., Equation 6 in Ryel et al., 2012): (1) This manuscript: $c_j$, the factor reducing soil-root conductance in the giving soil layer $j$ ; Ryel et al.:

15  max $(c_i, c_j)$, the larger one of the two factors in the receiving soil layer $i$ and the giving soil layer $j$; (2) In the denominator: This manuscript: $F_{root}(j)$, the root fraction in the giving soil layer $j$; Ryel et al.: This parameter can be $F_{root}(i)$ or $F_{root}(j)$, which depends on the comparison between soil moisture of the layer i and that of the layer j ; (3) This manuscript: It is not explicitly stated whether HR occurs during daytime in the modeling; Ryel et al.: It is stated that HR is "turned off" during daytime in the modeling of their study. These aspects can be explained in this manuscript.

20  R: The implementation is the same as described in Ryel et al. 2012. In the manuscript, $i$ and $j$ have been stated as receiving and giving soil layers, respectively, which means that the soil water potential $\varphi_j > \varphi_i$.  According to the Equation 7 in Ryel et al., 2012 (namely Equations (2) in the present study), $c_j = 1 \div [1 + (\varphi_j \div \varphi_{50})^b]$, with $b$=3.22 and $\varphi_{50}$ =-1 MPa, $\varphi_j > \varphi_i$ implies that $c_j > c_i$, so $c_j$ = max $(c_i, c_j)$.

In our simulation, we assumed that water was able to be transferred from giving layer $j$ to receive receiving layer $i$ only

25  under condition of $\varphi_j > \varphi_i$ and $\theta_j > \theta_i$. This assumption is only very slightly different from that in Reyl et al., 2002. We also set the HR off during the day. In the revised manuscript, we have added the following note: "$D$ is a switching factor, set to 1.0 during night and 0.0 during the day since, during day time, the transpiration-induced gradient of water potential within a plant continuum dictates a transport of water from roots to leaves".

In addition, if the definition of $C_{RT}$ here is same as that of Ryel et al., "of the entire active root system for water" needs to be

30  added after "soil-root conductance" in Page 4 Line 19.

R: It has been added as suggested.

2. Root distribution along the vertical direction in the soil column is an important input in the HR modeling. In Page 4 Line 20, it is mentioned that the root distribution is based on Zeng (2001). More information about root distribution can be

provided (e.g., how is the root distribution represented; Root-distribution differences between the eight AmeriFlux sites; etc.). This information can be helpful for the readers to understand the simulation results.

R: In the revised manuscript, we have added the following note:

"There are ten active soil layers in CLM, and a maximum depth of 3.8 m is used in this study (Table 3). The PFT-level root fraction $r_i$ in each soil layer is,

$$r_i = \begin{cases} 0.5 \cdot \left[ exp(-r_a z_{i-1}) + exp(-r_b z_{i-1}) - exp(-r_a z_i) + exp(-r_b z_i) \right] & for \quad 1 \leq i < 10 \\ 0.5 \cdot \left[ exp(-r_a z_{i-1}) + exp(-r_b z_{i-1}) \right] & for \qquad i = 10 \end{cases} \tag{1}$$

where $z_i$ is the depth at the bottom of soil layer $i$, and $z_0$ is zero. The PFT-dependent root distribution parameters $r_a$ and $r_b$ are adopted from Zeng (2001). From Equation (1), $r_i$ decreases exponentially with depth".

Technical Corrections

Page 1 Line 18: Add "land" between "on" and "surface water".

R: Changed as suggested.

Page 1 Line 19: Replace "it" with "the impact".

R: Changed as suggested.

Page 2 Lines 8-9: Remove "do so by".

Page 2 Line 9: Replace "incorporating" with "incorporate".

R: we think the original expression may reflect our thoughts better.

Page 2 Line 9: Add "e.g.," before "Ryel et al., 2002" (because some studies listed did not use the Ryel HR scheme, for example, Lee et al., 2005).

R: Changed as suggested.

Page 2 Line 24: Replace "CA" with "California".

R: Changed as suggested.

Page 2 Line 30: Replace "2m of soil" with "2-m soil layer".

R: Changed as suggested.

Page 3 Line 1: Replace "objective" with "objectives"; Replace "is" with "are".

R: Changed as suggested.

Page 3 Lines 1-2: Replace "modeling approach, the Ryel et al. (2002) approach," with "HR scheme (or model ?) (Ryel et al., 2002)".

R: we have changed "the Ryel et al. (2002) approach" to "the HR scheme of Ryel et al. (2002)" throughout the manuscript.

Page 3 Line 3: Add "land" between "on" and "surface water"; Replace "it" with "the impact".

R: Changed as suggested.

Page 3 Line 4: Rephrase "This is done through incorporating Ryel et al.'s (2002) simple empirical equation for HR flux" to be "For these objectives, we incorporated the Ryel HR scheme".

Page 3 Line 5: Add "(CLM4.5)" after "4.5"; Replace "applying" with "applied"; Remove "the" before "eight AmeriFlux sites".

R: we have rephrased the novelty of the present study in the revised manuscript, and correspondingly this sentence has been rephrased to "The modeling investigation is done through incorporating the HR scheme of Ryel et al. (2002) into the NCAR Community Land Model Version 4.5 (CLM4.5)".

Because we have described these eight sites, we would like to keep "the" before "eight AmeriFlux sites".

Page 3 Line 20: Replace "degrees C" with "°C".

R: Changed as suggested.

Page 4 Line 12: Replace "quantify" with "represent"; Replace "Ryel et al.'s (2002) equation" with "the Ryel HR scheme (Ryel et al., 2002)".

R: Changed as suggested.

Page 4 Lines 12-14: Rephrase "This equation has been widely used in HR modeling studies (Lee et al., 2005; Zheng and Wang, 2007; Wang, 2011; Li et al., 2012) and its variations (e.g. Yu and D'Odorico, 2015)." to be "Many HR modeling studies used this HR scheme (e.g., Zheng and Wang, 2007; Wang, 2011; Li et al., 2012) or its variations (e.g., Lee et al., 2005; Yu and D'Odorico, 2015).".

R: Changed as suggested.

Page 4 Line 25: Rephrase "The relationship between root hydraulic conductivities and soil moisture in equation (2) is similar to ..." to be "The relationship of Equation 2 is similar to ...". (because the description is not accurate)

R: In the revised manuscript, we have deleted this sentence to avoid confusion.

Page 4 Line 28: Replace "equation 1" with "Equation 1".

R: Changed as suggested.

Page 5 Line 1: Replace "MPb" with "MPa".

R: Changed as suggested.

Page 5 Line 2: Replace ""Ryel et al. 2002" equation" with "Ryel HR scheme".

R: Changed as suggested.

Page 5 Line 6: Add "Clapp and Hornberger" between "influences of" and ' "B" '.

R: Changed as suggested.

Page 5 Line 6: Replace "based on" with "in the".

R: Changed as suggested.

Page 5 Line 11: Replace ' "B"s ' with ' "B" values '; Replace "lower" with "deeper".

R: Changed as suggested.

Page 5 Lines 12-13: Rephrase "... identical parameter "B" tuned ..." to be "... the identical "B" value tuned ...".

R: Changed as suggested.

Page 5 Lines 16-17: Remove "for the sites".

R: Changed as suggested.

Page 5 Line 23: The meaning of "conservatively" is not clear.

R: "were used conservatively" has been changed to "were used more cautiously" in the revised manuscript.

Page 5 Lines 25-26: "... in surrounding portions of the signal trace ..." is not clear.

R: annotation of "larger oscillations beginning around day 180 in the 5 cm trace" has been added.

Page 5 Line 26: Add "existence of" before "HR".

R: Changed as suggested.

Page 6 Line 4: Replace "increment" with "soil layer".

R: Changed as suggested.

Page 6 Line 12: Add " – " after "the models".

R: Changed as suggested.

Page 6 Lines 20-21: Remove the comma " , "; Remove "US-SRM"; Add "at this site" after "... 90-100 cm depths".

R: Changed as suggested.

Page 6 Line 22: Replace "rain" with "soil water"; Replace "surface" with "shallow"; Rephrase "it could be delivered by percolation alone" to be "the percolation process".

R: we have revised this sentence to "when root systems redistribute the infiltrated rainwater from shallow to deep soils faster than it could be delivered by percolation alone". We think "infiltrated rainwater" is more suitable than "soil water" here; we are apt to keep "it could be delivered by percolation alone" unchanged, because two processes were referred here: hydraulic descent via roots and percolation through the soil.

Page 6 Line 31: The "hydrological effect" was "far lower" than what?

R: "(than wetter sites such as US-Wrc)" has been added here.

Page 7 Line 14: Add "flux" after "HR".

R: Changed as suggested.

Page 7 Line 16: Replace "... the magnitude of ..." with "... the magnitude in the ...". (to make this sentence more readable)

R: Changed as suggested.

Page 7 Line 19: The hydraulic descent magnitude (12 – 38 mm/day and 35 mm/day) seems to be very large. Please verify these values.

R: yes, we have checked. The hydraulic descent at the SRM site did have such magnitude at ecosystem scale.

Page 7 Line 25: "..., the maximum depth of HR-induced soil moisture increases ..." is not clear.

R: we have changed it to "the maximum depth at which the HR-induced soil moisture increases are identifiable".

Page 7 Lines 28-29: "... used soil suction to roughly control the magnitude of HR ..." is not clear.

R: we described it in section 2.3 as "At the three drier southern California sites (US-SCw, US-SCc, and US-SCd), $C_{RT}$ was further adjusted to relatively small values (0.05-0.1) to limit the hydraulic descent in order to reduce the model bias for soil water potential during dry periods. If $C_{RT} > 0.1$, the modeled soil water potential would be always higher than -1 MPa during

dry periods, which is not realistic for such dry sites". To keep consistence and remove confusion, we have changed "soil suction" to "soil water potential" here.

Page 8 Line 3: Remove "+HR" after "CLM4.5"; Replace "percent" with "percentage points".

R: Changed as suggested.

Page 8 Line 5: Add "those of" before "decreased soil moisture".

R: Changed as suggested.

Page 8 Line 8: Remove "at the six Southern California US-SC sites".

R: US-Wrc and US-SRM do not belong to the six Southern California US-SC sites, so we need to keep "at the six Southern California US-SC sites" unchanged here.

Page 8 Line 11: Add "in" before "2011".

R: Changed as suggested.

Page 8 Line 13: Replace "Ryel et al.'s (2002) HR model" with "the Ryel HR scheme".

R: Changed as suggested.

Page 8 Line 14: Remove "up to 1%"; Add " (up to 1%) " after "soil moisture".

R: Changed as suggested.

Page 8 Line 15: Remove "up to 4%"; Add " (up to 4%) " after "soil moisture".

R: Changed as suggested.

Page 8 Line 16: Remove "%"; Add " (in the unit of %) " after "soil moisture change".

R: Changed as suggested.

Page 8 Line 18: Switch "site" and "US-Wrc".

R: Changed as suggested.

Page 8 Line 19: Switch "HR" and "downward".

R: Changed as suggested.

Page 8 Lines 23-24: "... , a gradient in the depth and temporal extent of HR on modeled soil moisture was clear." needs to be rephrased.

R: it has been changed to "a gradient in the vertical and temporal effect of HR on modeled soil moisture was clear".

Page 8 Line 27: Add "range" after "depth"; Replace "influence of HR" with "HR influence".

R: Changed as suggested.

Page 8 Line 29: Add "H2O" between "mm" and "d-1". (to be consistent with the same unit in this paragraph)

R: Changed as suggested.

Page 8 Lines 30-32: It is not clear these results are from a specific year, or average values of multiple years?

R: "for all simulation years" has been added here.

Page 9 Line 4: Replace "varying" with "various".

R: Changed as suggested.

Page 9 Line 7: Remove "also improvement".

R: we are apt to keep it unchanged here.

Page 9 Line 9: Remove "(hourly) ".

R: Changed as suggested.

5 Page 9 Line 9: These results are average values of one year, or multiple years?

R: "for all simulation years" has been added here.

Page 9 Line 10: Replace "tends" with "tended".

R: Changed as suggested.

Page 9 Line 10: Remove "observed mid-day". (These words are redundant.)

10 R: Changed as suggested.

Page 9 Lines 10-11: Fig. 6 shows that, for most circumstances, the modeled high ET values from CLM4.5+HR are closer to the observations, as compared to those from CLM4.5. This point can be put forward around here.

R: the following sentence has been added "Under most circumstances, the simulated ET peaks in CLM4.5+HR are closer to observations than those in CLM4.5".

15 Page 9 Lines 17-18: "... was subsequently taken up from deeper layers by plants during transpiration." – But soil water in deep layers could also be transferred to the shallow soil via the upward HR process when plant transpiration rate is low.

R: Hydraulic lift might occur later, but the lifted soil water to shallow soil layers was also mainly consumed by transpiration. This sentence has been changed to "… was eventually consumed by plants during subsequent transpiration".

20 Page 9 Line 21: Move "(0.47 mm H2O d-1)" to after "US-SCf site"; Remove "corresponding ET increase is".

R: Changed as suggested.

Page 9 Line 22: Add "site" after "US-SRM".

R: Changed as suggested.

Page 9 Line 25: Replace "Ratio" with "ratio".

25 R: Changed as suggested.

Page 9 Line 32: Rephrase "synthesizes" to be "reflects the integration of".

R: we think "synthesize" is a suitable word here, and it is concise compared with "reflects the integration of".

Page 10 Line 5: Rectify "hear" to be "heat".

R: Changed as suggested.

30 Page 10 Line 8: "... beyond those studied here" is not clear.

R: this sentence has been changed to "also commonly measured at other field sites".

Page 10 Line 10-11: Rephrase "... HR-induced increases in ET are maximal at sites with midrange soil moistures" to be " ... HR-induced large increases in ET primarily occur at sites with mid-range soil moistures". (because in Fig. 8, for sites of moderate soil moisture, some HR induced increases of ET are small)

R: it has been changed to "it is clear that maximal HR-induced increases in ET primarily occur at sites with mid-range soil moistures".

Page 10 Line 14: Replace ' "Ryel et al. 2002" equation ' with 'Ryel HR scheme'.

R: Changed as suggested.

Page 10 Line 17: "at the height of HR" is not clear.

Page 10 Line 17: Add "during the periods with high HR flux". (because in Fig. S10, hydraulic lift magnitude could double when HR flux is moderate or low)

R: "at the height of HR" has been changed to "during the periods with high HR flux".

Page 10 Line 27: Add "as" after "considered".

R: Changed as suggested.

Page 11 Line 7: Replace "yr" with "year".

R: Changed as suggested.

Page 11 Line 13: Replace "dominated" with "primarily caused".

R: because both hydraulic lift and hydraulic descent (although very minor) exist during dry season, we are apt to keep "dominated" unchanged.

Page 11 Line 14: Replace "H2O" with "$H_2O$".

R: Changed as suggested.

Page 11 Line 17: Add "soil" before "water content".

R: Changed as suggested.

Page 11 Line 20: Replace "is" with "was".

R: Changed as suggested.

Page 11 Line 25: Add "missing representation of" before "hydraulic lift".

R: Changed as suggested.

Page 11 Line 31: Remove "the monitored".

R: Changed as suggested.

Page 12 Line 2: Rephrase ", including at sites modeled here" to be "including those conducted at sites of this study".

R: we changed this to "including at sites studied here".

Page 12 Line 3: Replace "contributed" with "supplied"; Replace "to" with "of"; Rephrase "upper 2 m of the soil" to be "top 2-m soil layer".

R: Changed as suggested.

Page 12 Line 5: Replace "similar" with "comparable". ( The meaning of 28% is different from that of 32%: 28% = HL / ET from the top 2-m layer; 32% = HL / ET from the whole soil column. )

R: Changed as suggested.

Page 12 Lines 6-7: Add "transpiration of the" after "estimated"; Remove "transpiration" before "in 2004".

R: Changed as suggested.

Page 12 Line 21: Replace "are" with "were".

R: Changed as suggested.

Page 12 Line 28: Replace "idea" with "hypothesis" or "assumption".

5   R: Changed as suggested.

Page 13 Line 3: The reference "Quijano and Kumar, 2015" is missing in the reference list.

R: this reference has been added to the reference list.

Page 13 Lines 8-10: These hydraulic lift results are maximum values in a time period? Or mean values of some periods?

R: "; Table 6" has been added here, and we have also added "for all simulation years" to the caption of Table 6.

10   Page 13 Line 14-16: These ET increases are maximum values in a time period? Or mean values of some periods?

R: "; Table 6" has been added here, and we have also added "for all simulation years" to the caption of Table 6.

Page 13 Line 23: Replace "Ryel et al. 2002 equation" with "Ryel HR scheme".

R: Changed as suggested.

Page 13 Line 25: Add "during the periods with high HR flux" after "hydraulic lift".

15   R: Changed as suggested.

Page 13 Lines 25-26: Add "larger change of" before "hydraulic descent"; Remove "was more sensitive".

R: we think both descriptions are OK, and are apt to keep the original statement unchanged.

Page 13 Lines 29-31: "HR has been confirmed in many ecosystems" does not convincingly support the argument that "HR should be included for all ecosystems". Need some revising here.

20   R: "HR has been confirmed in many ecosystems" has been changed to "HR has been confirmed in various ecosystems" here.

Page 13 Line 32: Replace "a large number of" with "eight". (Eight is not a large number.)

R: Changed as suggested.

Page 18 Lines 3, 6: Replace "Ryel et al.'s 2002 equation describing HR" with "the Ryel HR scheme".

R: Changed as suggested.

25   Page 19: Columns 1 and 2: it is better to align left.

R: Changed as suggested.

Page 20: Column 5 Row 3: Add "m" after "0.25-5".

R: Changed as suggested.

Page 22 Line 1: Replace '"Ryel et al. 2002" HR model ' with "Ryel HR scheme".

30   R: Changed as suggested.

Page 22 Line 2: Add "of the entire active root system for water" after the first "conductance", if the definition of CRT is same as that in Ryel et al. (2002).

R: Changed as suggested.

Page 22 Line 3: Remove the quotation marks " " around b .

R: Changed as suggested.

Page 23: Need to specify the results of Table 5 are mean values of one year or mean values of multiple years.

R: "for all simulation years" has been added in the caption.

Page 23: The last column: these values need to be multiplied by 100.

R: Changed as suggested.

Page 24: The 4th line from the bottom: Replace "soil moistures" with "soil moisture data".

R: Changed as suggested.

Page 24: The 2nd line from the bottom: Remove "RMSE for". (They are redundant.)

R: Changed as suggested.

Page 24: The last line: Rephrase "improved model fit to data" to be "a better fit between model results and observed data".

R: we think both descriptions are OK, and are apt to keep the original statement unchanged.

Page 25: The 2nd Line from the bottom: Replace "at 0-30 cm depth" with "at different depths in the top 30-cm soil layer".

R: The monitoring depths of all probes at each southern California site are all 0-30 cm, so we keep "at 0-30 cm depth" unchanged here.

Page 25: The last two lines: "Results were shown for the last two years only for the above 30 cm depth at US-SRM site for clarity." is not clear. This sentence seems to mean that: "For the top 30-cm soil layer at the US-SRM site, only the results of the last year (Year 2012) were shown for clarity."

R: it has been changed to "Results were shown for the last year only (2012) for the top 30 cm depths at US-SRM site for clarity".

Page 25: The last line: "The rectangular box" is missing in the figure.

R: it has been added in the revised manuscript.

Page 27: The last line: Remove the first "soil moisture".

R: Changed as suggested.

Page 28: The units are missing in Fig. 4.

R: "(unit: %)" has been added.

Page 29 Lines 1 and 2: Remove "(mm)" in the legend.

R: Changed as suggested.

Page 30: In the ET plots, the meaning of the thin vertical lines is not described.

R: the following note has been added in the caption "The thin vertical lines represent one standard deviation of simulation with HR".

Page 30: The 2nd line from the bottom: "dry season and wet season" are from one year or multiple years.

R: "for all simulation years" has been added in the caption.

Page 31: The results of Year 2004 at the US-SRM site are contrary to the other results, which can be discussed in the text.

R: The simulation at the SRM site started with a period with significant hydraulic descent (January in 2004), which resulted in lower simulated soil moisture with HR than that without HR in shallow soil layers (e.g., at depth 0-30 cm) before Julian day 180, although hydraulic lift occurred during days 120-180. Lower simulated soil moisture resulted in lower/smaller simulated ET, as shown in results of 2004 in Fig. 5 and 7. So the lower simulated ET with HR during the dry period (around 120-180) than that without HR, could be seen as the "spinning up" process of the model. Considering the substantial amount of information in the manuscript, we have not added explanations about this phenomenon in order to avoid confusions in the revised manuscript. Instead, in the revised manuscript, we have deleted the results for 2004 for the SRM site in Figs. 5 and 7.

Page 33: Replace "$\varphi 50$" with "$\varphi 50$".

R: "$\varphi 50$" has been changed to "$- \varphi 50$".

Page 33: Units of CRT and $\varphi 50$ are missing.

R: Units have been added in the revised manuscript.

Page 33: The last two lines: Replace "Ryel et al.'s 2002 equation describing HR" with "the Ryel HR scheme".

R: Changed as suggested.

[revised manuscript text omitted]